# Associations of four biological age markers with child development: A multi-omic analysis in the European HELIX cohort

Oliver Robinson[1,2]*, ChungHo E Lau[1], Sungyeon Joo[1], Sandra Andrusaityte[3], Eva Borras[4,5], Paula de Prado-Bert[5,6,7], Lida Chatzi[8], Hector C Keun[9,10], Regina Grazuleviciene[3], Kristine B Gutzkow[11], Lea Maitre[5,6,7], Dries S Martens[12], Eduard Sabido[4,5], Valérie Siroux[13], Jose Urquiza[5,6,7], Marina Vafeiadi[14], John Wright[15], Tim S Nawrot[12], Mariona Bustamante[5,6,7], Martine Vrijheid[5,6,7]

[1]Medical Research Council Centre for Environment and Health, Imperial College London, London, United Kingdom; [2]Mohn Centre for Children's Health and Well-being, School of Public Health, Imperial College London, London, United Kingdom; [3]Department of Environmental Science, Vytautas Magnus University, Kaunas, Lithuania; [4]Center for Genomics Regulation, Barcelona Institute of Science and Technology, Barcelona, Spain; [5]Universitat Pompeu Fabra (UPF), Barcelona, Spain; [6]Institute for Global Health (ISGlobal), Barcelona, Spain; [7]CIBER Epidemiologa y Salud Pública (CIBERESP), Madrid, Spain; [8]Department of Preventive Medicine, Keck School of Medicine, University of Southern California, Los Angeles, United States; [9]Division of Systems Medicine, Department of Metabolism, Digestion and Reproduction, Imperial College London, London, United Kingdom; [10]Cancer Metabolism & Systems Toxicology Group, Division of Cancer, Department of Surgery & Cancer; Imperial College London, London, United Kingdom; [11]Division of Climate and Environmental Health, Norwegian Institute of Public Health, Oslo, Norway; [12]Centre for Environmental Sciences, Hasselt University, Hasselt, Belgium; [13]University Grenoble Alpes, Inserm U 1209, CNRS UMR 5309, Team of environmental epidemiology applied to the development and respiratory health, Institute for Advanced Biosciences, 38000 Grenoble, France; [14]Department of Social Medicine, School of Medicine, University of Crete, Heraklion, Crete, Greece; [15]Bradford Institute for Health Research, Bradford Teaching Hospitals NHS Foundation Trust, Bradford, United Kingdom

*For correspondence:
o.robinson@imperial.ac.uk

Competing interest: The authors declare that no competing interests exist.

## Abstract

**Background:** While biological age in adults is often understood as representing general health and resilience, the conceptual interpretation of accelerated biological age in children and its relationship to development remains unclear. We aimed to clarify the relationship of accelerated biological age, assessed through two established biological age indicators, telomere length and DNA methylation age, and two novel candidate biological age indicators, to child developmental outcomes, including growth and adiposity, cognition, behavior, lung function and the onset of puberty, among European school-age children participating in the HELIX exposome cohort.

**Methods:** The study population included up to 1173 children, aged between 5 and 12 years, from study centres in the UK, France, Spain, Norway, Lithuania, and Greece. Telomere length

was measured through qPCR, blood DNA methylation, and gene expression was measured using microarray, and proteins and metabolites were measured by a range of targeted assays. DNA methylation age was assessed using Horvath's skin and blood clock, while novel blood transcriptome and 'immunometabolic' (based on plasma proteins and urinary and serum metabolites) clocks were derived and tested in a subset of children assessed six months after the main follow-up visit. Associations between biological age indicators with child developmental measures as well as health risk factors were estimated using linear regression, adjusted for chronological age, sex, ethnicity, and study centre. The clock derived markers were expressed as Δ age (i.e. predicted minus chronological age).

**Results:** Transcriptome and immunometabolic clocks predicted chronological age well in the test set ($r$=0.93 and $r$=0.84 respectively). Generally, weak correlations were observed, after adjustment for chronological age, between the biological age indicators.

Among associations with health risk factors, higher birthweight was associated with greater immunometabolic Δ age, smoke exposure with greater DNA methylation Δ age, and high family affluence with longer telomere length.

Among associations with child developmental measures, all biological age markers were associated with greater BMI and fat mass, and all markers except telomere length were associated with greater height, at least at nominal significance ($p<0.05$). Immunometabolic Δ age was associated with better working memory ($p=4$ e–3) and reduced inattentiveness ($p=4$ e–4), while DNA methylation Δ age was associated with greater inattentiveness ($p=0.03$) and poorer externalizing behaviors ($p=0.01$). Shorter telomere length was also associated with poorer externalizing behaviors ($p=0.03$).

**Conclusions:** In children, as in adults, biological aging appears to be a multi-faceted process and adiposity is an important correlate of accelerated biological aging. Patterns of associations suggested that accelerated immunometabolic age may be beneficial for some aspects of child development while accelerated DNA methylation age and telomere attrition may reflect early detrimental aspects of biological aging, apparent even in children.

**Funding:** UK Research and Innovation (MR/S03532X/1); European Commission (grant agreement numbers: 308333; 874583).

## Editor's evaluation

This is an important study that examined multiple biological age measures in children, which has been lacking in literature. The findings of this study provided convincing evidence to interpret and understand the aging and developmental processes in children.

## Introduction

The field of geroscience proposes that biological aging, a set of interrelated molecular and cellular changes associated with aging, drives the physiological deterioration that is the root of multiple age-related health conditions (*Sierra, 2016*). Understanding the process of biological aging and developing markers to accurately assess biological age in individuals, holds great promise for public health and biomedical research in general to develop interventions, even in childhood and early life, that slow physiological decline and reduce the risk of chronic disease and disability in later life.

Telomere length, which shortens with age, is one of the most widely applied biological age markers primarily as it directly assesses a primary Hallmark of Aging (*López-Otín et al., 2013*; *Jylhävä et al., 2017*). More recently, high-throughput 'omic' methods, which provide simultaneous quantification of thousands of epigenetic marks, transcripts, proteins, and metabolites, have been used to develop 'biological clocks' that provide a global measure of changes with age at the molecular level (*Rutledge et al., 2022*). While biological clocks have been primarily trained on chronological age, 'age acceleration,' commonly defined as the difference between clock-predicted age and chronological age, has been associated with age-related phenotypes and mortality (*Hertel et al., 2016*; *Perna et al., 2016*; *Marioni et al., 2015*; *Peters et al., 2015*; *Lehallier et al., 2019*; *van den Akker et al., 2020*; *Macdonald-Dunlop et al., 2022*), indicating their utility as biological age markers. DNA methylation-based clocks, such as the clock of *Horvath, 2013*, have been extensively applied in large-scale studies and remain a research field under active development, with 'second generation' clocks further

**eLife digest** Although age is generally measured by the number of years since birth, many factors contribute to the rate at which a person physically ages. In adults, linking these measurements to age gives a measure of overall health and resilience. This 'biological age' offers a better prediction of remaining life and disease risk than the number of years lived.

Multiple factors can be used to calculate biological age, such as measuring the length of telomeres – protective caps on the end of chromosomes – which shorten as people age. The rate at which they shorten can give an indication of how quickly someone is ageing. Researchers can also study epigenetic factors: these mechanisms lead to certain genes being switched on or off, and they can be combined into a 'epigenetic clock' to assess biological age. However, compared with adults, the relationship between biological age and child health and developmental maturity is less well understood.

Robinson et al. studied 1,173 school-aged children from six European countries, measuring telomere length, epigenetic factors and other biological indicators related to metabolism and the immune system. The relationships between these factors and an array of child developmental measures such as height, weight, behaviour and the age of onset of puberty were established. The findings showed that biological age indicators are only weakly linked to each other in children. Despite this, biological age was related to greater amount of body fat across all tested indicators – which is also associated with biological age in adults and is an important determinant of lifespan.

Among several observed effects on development, analysis found that shorter telomere length and older epigenetic age were associated with greater behavioural problems, suggesting they may be detrimental to child development. On the other hand, a greater age due to metabolic and immune related changes was associated with greater cognitive and behavioural maturity. Environmental factors were also linked to biological ageing, with children exposed to smoking in their homes or while their mother was pregnant displaying an older epigenetic age.

Robinson et al. showed that biological ageing in children is multifaceted and can have both beneficial and harmful impacts on development. This knowledge is important for identifying early life risk factors that might influence healthy ageing in later life. Future work will help researchers to understand these complex interactions and the long-term consequences for health and well-being.

incorporating clinical biomarker and mortality information to improve their clinical utility (*Lu et al., 2019*; *Belsky et al., 2020*). Further clocks have been developed using transcriptome (*Peters et al., 2015*), metabolome (*Robinson et al., 2020*), and proteome (*Lehallier et al., 2019*) data, including those that specifically target immune-system-related proteins (*Sayed et al., 2021*). Generally, clocks have been found to be only weakly correlated with each other, suggesting that each clock captures different facets of biological aging (*Belsky et al., 2018*; *Jansen et al., 2021*).

While biological age in adults is intuitively understood as an overall indicator of general health and resilience, the conceptual interpretation of biological age acceleration in children is much less clear. Child development and aging may at first be considered opposing processes, representing growth and decay respectively. However, various related theoretical frameworks link the two processes: Under the developmental origin of health and disease hypothesis, the early life environment is a key determinant of ageing trajectories and disease risk in later life. Life-course models of ageing, supported by measures of physical and cognitive capability, view the childhood developmental phase as key to building up 'biological capital' and to determining how long capabilities and disease risk remain above critical thresholds in later years following the gradual decline phase of adult life (*Kuh, 2007*). Horvath's DNA methylation clock is currently the only clock trained to predict age throughout the lifespan, and many of the clock's CpGs are in genomic regions known to regulate development and differentiation (*Horvath, 2013*). However, unlike life-course models of physical function, the level of DNA methylation at the clock's CpGs changes in a predictable, unidirectional manner throughout the life-course, albeit at a much faster rate during childhood. This continuous molecular readout suggests that processes directing development are at least indirectly related to the detrimental processes in later-life and is consistent with quasi-programmed theories of aging such as antagonistic pleiotropy (*Emery Thompson, 2022*), whereby molecular functions that promote development, inadvertently lead to aging in later life (*Horvath and Raj, 2018*). Therefore, some authors have suggested that DNA

methylation-based age acceleration may be beneficial during childhood (*Horvath and Raj, 2018*; *McEwen et al., 2020*), reflecting greater physical maturity and build-up of biological capital.

Biological aging is conceived as a continuous balance of cellular damage, caused by both extrinsic environmental factors and by normal physiological processes, and resiliency mechanisms that protect against and compensate for this damage (*Ferrucci et al., 2020*). An alternative 'wear and tear' model would view cellular damage to occur continuously from birth and, since the epigenetic clock has been proposed to reflect the epigenomic maintenance system, a resiliency mechanism, DNA methylation age acceleration in children may, as in adults, represent a greater accumulation of epigenetic instability and, therefore, reduced biological capital. However, so far only a handful of studies have examined associations with developmental maturity in children (*Binder et al., 2018*; *Simpkin et al., 2017*; *Suarez et al., 2018*). Telomere length attrition is more rapid in early childhood during rapid somatic growth and more gradual in adulthood, with those with a shorter telomere length in childhood maintaining a lower telomere length into adulthood (*Martens et al., 2021*). While telomere length may serve as both a mitotic-clock and as a mediator of cellular stress (*Coimbra et al., 2017*), the associations reported between environmental stress in childhood and shorter telomere length suggest it reflects early-life cellular damage that may be carried into adulthood.

Little is known regarding the interpretation of biological age in children assessed at the transcriptome, proteomic and metabolomic levels, since few biological clocks are available for this age range using these data. To the best of our knowledge, only the study of *Giallourou et al., 2020* has applied metabolomic data to provide a multivariate model of age in children, finding that growth-constrained infants lag in their metabolic maturity relative to their healthier peers. It is possible that biological clocks constructed using these data, particularly proteomics and metabolomics, support the life-course aging framework, where age acceleration in children represents a buildup of biological capital, since they are closer to the phenotype than the DNA-based epigenetic clocks and telomere length.

We hypothesized that age acceleration would be associated with child development. To test this and assess whether age acceleration is associated with beneficial or detrimental effects on child development, we have performed a comparative analysis of two established and two candidate

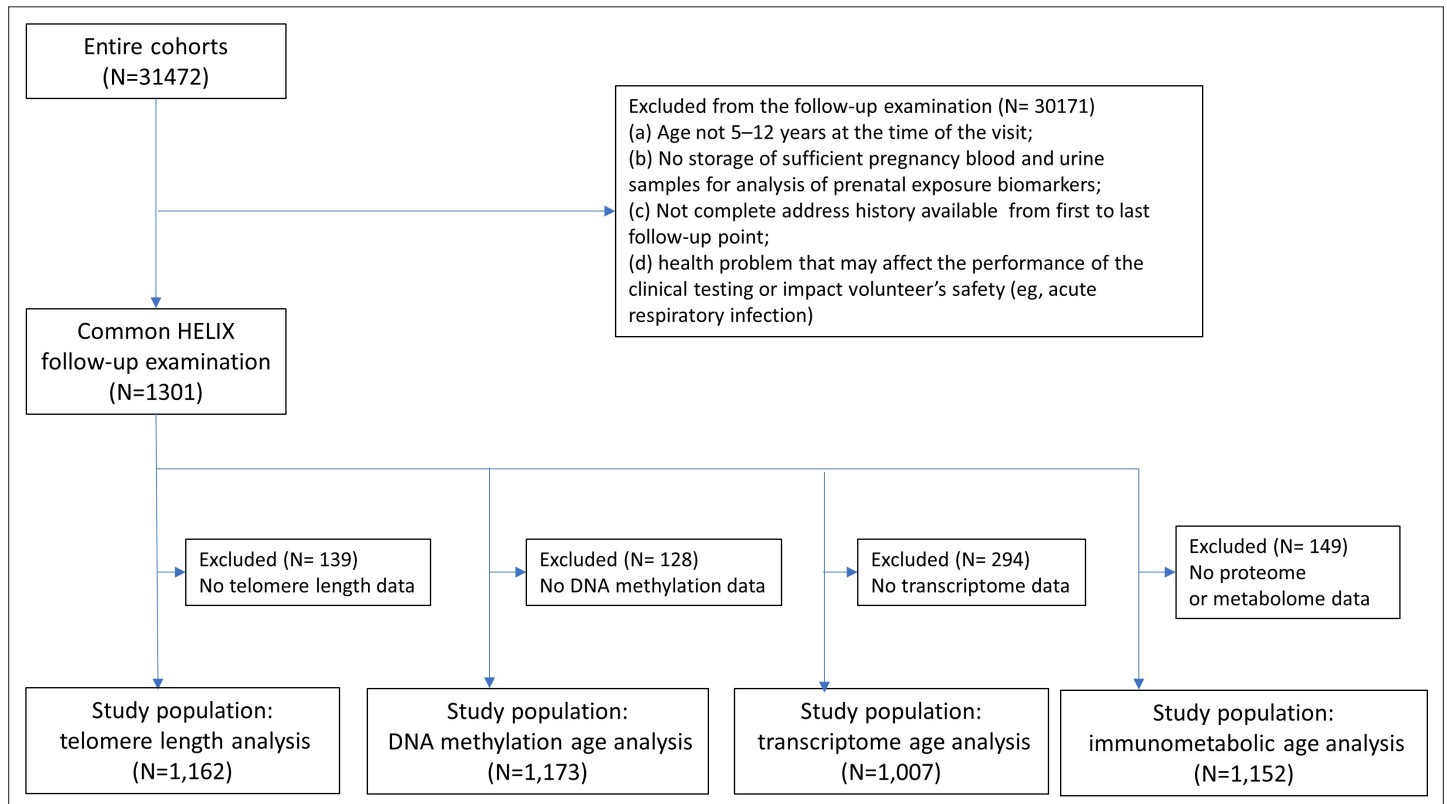

**Figure 1.** Participant flowchart. See *Supplementary file 1* for details on quality control of molecular data at sample and feature levels.

assessments of biological age within the pan-European Human Early Life Exposome (HELIX) cohort of children aged between 5 and 12 years. We systematically compared associations with developmental endpoints, including growth and adiposity, cognition, behavior, lung function and pubertal development, and common health risk factors, for telomere length, DNA methylation age, and two newly derived clocks: transcriptome age and immunometabolic age. Through this analysis, we aimed to clarify the interpretation of age acceleration in children, and more broadly develop new biological markers of overall developmental staging in children.

## Materials and methods
### Study population
This study population included children recruited into the European population-based HELIX exposome cohort (*Maitre et al., 2018*; *Vrijheid et al., 2014*), which was based on six on-going longitudinal population-based birth cohorts established in six countries across different parts of Europe (Born in Bradford [BiB; UK] *Wright et al., 2013*), Étude des Déterminants Pré et Postnatals du Développement et de la Santé de l'Enfant [EDEN; France] (*Heude et al., 2016*), Infancia y Medio Ambiente [INMA; Spain] (*Guxens et al., 2012*), Kaunas Cohort [KANC; Lithuania] (*Grazuleviciene et al., 2009*), Norwegian Mother, Father and Child Cohort Study [MoBa; Norway] (*Magnus et al., 2016*), and Mother-Child Cohort in Crete [RHEA; Greece] (*Chatzi et al., 2017*) covering singleton deliveries from 2003–2008. All children participated in a harmonized 'HELIX subcohort' clinical examination in their respective study centres during 2014–2015, where biological samples were collected. A subset of children (from all study centres apart from MoBa), attended a second clinical examination, as part of the 'HELIX panel study' approximately 6 months after the first 'HELIX subcohort' examination, where a similar suite of biological samples was collected. A full description of the HELIX follow-up methods and study population, including eligibility criteria and sample size calculations are available in *Agier et al., 2019*; *Quanjer et al., 2012*. In the current study, we included all children with available molecular data (*Figure 1*).

Prior to the start of HELIX, all six cohorts had undergone the required evaluation by national ethics committees and obtained all the required permissions for their cohort recruitment and follow-up visits. Each cohort also confirmed that relevant informed consent and approval were in place for the secondary use of data from pre-existing data. The work in HELIX was covered by new ethical approvals in each country and at enrolment in the new follow-up, participants were asked to sign a new informed consent form. Additionally, the current study was approved by the Imperial College Research Ethics Committee (Reference: 19IC5567).

### Biological sample collection and processing
Blood was collected at the end of the clinical examination of the child to ensure an approximate 3 hr (median=3.5 hr, SD=1.1 hr) fasting time since the last meal. Blood samples were collected using a 'butterfly' vacuum clip and local anesthetic and processed into a variety of sample matrices, including plasma, whole blood for RNA extraction (Tempus tubes - Life Technologies, USA), red cells, and buffy coat for DNA extraction. These samples were frozen at –80 °C under optimized and standardized procedures until analysis.

DNA was obtained from children's peripheral blood (buffy coat) collected in EDTA tubes. DNA was extracted using the Chemagen kit (Perkin Elmer, USA) in batches of 12 samples within each cohort. DNA concentration was determined in a Nanodrop 1000 UV-Vis Spectrophotometer (Thermo Fisher Scientific, USA) and also with Quant-iTTM PicoGreen dsDNA Assay Kit (Life Technologies, USA). DNA extraction was repeated in around 8% of the blood samples as the DNA quantity or quality of the first extraction was low. Less than 1.5% of the samples were finally excluded due to low quality.

RNA was extracted from whole blood samples collected in Tempus tubes (Thermo Fisher Scientific, USA) using MagMAX for Stabilized Blood Tubes RNA Isolation Kit. The quality of RNA was evaluated with a 2100 Bioanalyzer (Agilent Technologies, USA), and the concentration with a NanoDrop 1000 UV-Vis Spectrophotometer. Samples classified as good RNA quality (78.67%) had a similar RNA pattern at visual inspection in the Bioanalyzer, an RNA Integrity Number (RIN)>5, and a concentration >10 ng/ul. Mean values (standard deviation, SD) for the RIN, concentration (ng/ul), Nanodrop 260/280 ratio and Nanodrop 260/230 ratio were: 7.05 (0.72), 109.07 (57.63), 2.15 (0.16), and 0.61 (0.41).

During the clinical examination, two spot urine samples (one before bedtime and one first morning void) were brought by the participants to the research centre in cool packs and stored at 4 °C until processing. Urine samples of the night before the visit and the first-morning void on the day of the visit were combined to provide two urine samples, representing the last night-time and first morning voids, which were collected on the evening and morning before the clinical examination and were subsequently pooled to generate a more representative sample of the last 24 hr for metabolomic analysis (N=1107). Either the night-time void (N=37) or morning void (N=48) sample was analyzed in cases where a pooled sample was missing (*Maitre et al., 2018*).

## Telomere length measurement

Blood average relative telomere length was measured by a modified qPCR protocol as described previously (*Cawthon, 2009*). Telomere and single copy-gene reaction mixture and PCR cycles used can be found in *Martens et al., 2016*. All measurements were performed in triplicate on a 7900HT Fast Real-Time PCR System (Applied Biosystems) in a 384-well format. On each run, a six-point serial dilution of pooled DNA was run to assess PCR efficiency as well as eight inter-run calibrators to account for the inter-run variability. Relative telomere lengths were calculated using qBase software (Biogazelle, Zwijnaarde, Belgium) and were expressed as the ratio of telomere copy number to single-copy gene number (T/S) relative to the average T/S ratio of the entire sample set. We achieved CV's within triplicates of the telomere runs, single-copy gene runs, and T/S ratios of 0.84%, 0.43%, and 6.4%, respectively.

## DNA methylation

Blood DNA methylation was assessed with the Infinium HumanMethylatio450 beadchip (Illumina, USA) at the University of Santiago de Compostela – Spanish National Genotyping Center (CeGen-USC) (Spain). 700 ng of DNA were bisulfite-converted using the EZ 96-DNA kit (Zymo Research, USA) following the manufacturer's standard protocol. All samples of the study were randomized considering sex and cohort. In addition, each plate contained a HapMap control sample and 24 HELIX inter-plate duplicates were included.

After an initial inspection of the quality of the methylation data with the MethylAid package (*van Iterson et al., 2014*), probes with a call rate <95% based on a detection p-value of 1e−16, and samples with a call rate <98% were removed (*Lehne et al., 2015*). Samples with discordant sex were eliminated from the study as well as duplicates with inconsistent genotypes and samples with inconsistent genotypes with respect to existing genome-wide genotyping array data. Methylation data were normalized using the functional normalization method with prior background correction with Noob (*Fortin et al., 2014*). Then, some probes were filtered out: control probes, probes to detect single nucleotide polymorphisms (SNPs), probes to detect methylation in non-CpG sites, probes located in sexual chromosomes, cross hybridizing probes (*Chen et al., 2013*), probes containing an SNP at any position of the sequence with a minor allele frequency (MAF) >5% and probes with an SNP at the CpG site or at the single base extension (SBE) at any MAF in the combined population from 1000 Genomes Project (*Supplementary file 1*). Batch effect (slide) was corrected using the ComBat R package (*Johnson et al., 2007*). CpGs were annotated with the IlluminaHumanMethylation450kanno.ilmn12.hg19 R package (*Hansen KD, 2012*).

## Transcriptome analysis

Gene expression was assessed using the GeneChip Human Transcriptome Array 2.0 (HTA 2.0) from Affymetrix (USA) at the University of Santiago de Compostela (USC) (Spain). Briefly, RNA samples were concentrated or evaporated in order to reach the required RNA input concentration (200 ng of total RNA). Amplified and biotinylated sense-strand DNA targets were generated from total RNA. Microarrays were hybridized according to the Affymetrix recommendations using the Affymetrix labeling and hybridization kits. All samples were randomized within each batch considering sex and cohort. Two different types of control RNA samples (HeLa and FirstChoice Human Brain Reference RNA (Thermo Fisher Scientific, USA)) were included in each batch, but they were hybridized only in the first batches.

Raw data were extracted with the Affymetrix AGCC software and normalized with the GCCN (SST-RMA) algorithm at the gene level (http://tools.thermofisher.com/content/sfs/brochures/ sst_gccn_whitepaper.pdf). Annotation of transcripts clusters (TCs) to genes was done with the Affymetrix

Expression Console software using the HTA-2_0 Transcript Cluster Annotations Release na36 (hg19). A transcript cluster is defined as a group of one or more probes covering a region of the genome reflecting all the exonic transcription evidence known for the region and corresponding to a known or putative gene. Four samples with discordant sex were detected with the MassiR R package (*Buckberry et al., 2014*) and excluded. Control probes, and TCs in sexual chromosomes and without chromosome information were filtered out. Batch effect (slide) was corrected using the ComBat R package (*Johnson et al., 2007*). To determine the TC call rate, 10 constitutive or best probes based on probe scoring and cross-hybridization potential were selected per TC. Probe Detection Above Background (DABG) p-values were computed based on the rank order against the background probe set intensities. Probe level p-values were combined into a TC level p-value using the Fisher equation. TCs with a DABG p-value <0.05 were defined as detected. Three samples with low call rate (<40%) as well as TCs with a call rate <1% were excluded from the dataset (*Supplementary file 1*). Gene expression values were log2 transformed.

## Proteome analysis

Plasma protein levels were assessed using the antibody-based multiplexed platform from Luminex. Three kits targeting 43 unique candidate proteins were selected (Thermo Fisher Scientific, USA): Cytokines 30-plex (Catalog Number (CN): LHC6003M), Apolipoprotein 5-plex (CN: LHP0001M) and Adipokine 15-plex (CN: LHC0017M).

All samples were randomized and blocked by the cohort prior to measurement. For quantification, an eight-point calibration curve per plate was performed with protein standards provided in the Luminex kit and following procedures described by the vendor. Commercial heat-inactivated, sterile-filtered plasma from human male AB plasma (Sigma-Aldrich, USA) was used as constant samples to control for intra- and inter-plate variability. Four control samples were added per plate. All samples, including controls, were diluted ½ for the 30-plex kit, ¼ for the 15-plex kit, and 1/2500 for the 5-plex kit.

Raw intensities obtained with the xMAP and Luminex system for each plasma sample were converted to pg/ml using the calculated standard curves of each plate and accounting for the dilutions made prior to measurement. The percentages of coefficients of variation (CV%) for each protein by plate ranged from 3–6%. The limit of detection (LOD) and the lower and upper limit of quantification (LOQ1 and LOQ2, respectively) were estimated by plate, and then averaged. Only proteins with >30% of measurements in the linear range of quantification were kept in the database and the others were removed. Seven proteins were measured twice (in two different multiplex kits). We kept the measure with higher quality. The 36 proteins that passed the quality control criteria mentioned above were log2 transformed (*Vives-Usano et al., 2020*). Then, the plate batch effect was corrected by subtracting the plate-specific average for each protein minus the overall average of all plates for that protein. After that, values below the LOQ1 and above the LOQ2 were imputed using a truncated normal distribution implemented in the truncdist R package (*Nadarajah and Kotz, 2006*). Twenty samples were excluded due to having ten or more proteins out of the linear range of quantification (*Supplementary file 1*).

## Metabolomic analysis

The AbsoluteIDQTM p180 kit was chosen for serum analysis as it is a standardized, targeted LC-MS/MS assay, widely used for large-scale epidemiology studies and its inter-laboratory reproducibility has been demonstrated by several independent laboratories (*Siskos et al., 2017*). Serum samples were quantified using the AbsoluteIDQTM p180 kit following the manufacturer's protocol (User Manual UM_p180_AB_SCIEX_9, Biocrates Life Sciences AG) using LC-MS/MS; an Agilent HPLC 1100 liquid chromatography coupled to a SCIEX QTRAP 6500 triple quadrupole mass spectrometer. A full description of the HELIX metabolomics methods and data can be found elsewhere (*Lau et al., 2018*).

Briefly, the kit allows for the targeted analysis of 188 metabolites in the classes of amino acids, biogenic amines, acylcarnitines, glycerophospholipids, sphingolipids, and the sum of hexoses, covering a wide range of analytes and metabolic pathways in one targeted assay. The kit consists of a single sample processing procedure, with two separate analytical runs, a combination of liquid chromatography (LC) and flow injection analysis (FIA) coupled to tandem mass spectrometry (MS/MS). Isotopically labeled and chemically homologous internal standards were used for quantification.

The AbsoluteIDQ p180 data of serum samples were acquired in 18 batches. Every analytical batch, in a 96-well plate format, included up to 76 randomized cohort samples. Also in every analytical batch, three sets of quality control samples were included, the NIST SRM 1950 plasma reference material (in four replicates), a commercially available serum QC material (CQC in two replicates, SeraLab, S-123-M-27485), and the QCs provided by the manufacturer in three concentration levels. The NIST SRM 1950 reference was used as the main quality control sample for the LC-MS/MS analysis. Coefficients of variation (CVs) for each metabolite were calculated based on the NIST SRM 1950 and also the limits of detection (LODs) were also used to assess the analytical performance of individual metabolites. Metabolite exclusion was based on a metabolite variable meeting two conditions: (1) CV of over 30% and (2) over 30% of the data are below LOD. Eleven out of the 188 serum metabolites detected were excluded as a result, leaving 177 serum metabolites to be used for further statistical analysis (*Supplementary file 1*). The mean coefficient of variation across the 177 LC-MS/MS detected serum metabolites was 16%. We also excluded one HELIX sample, which was hemolyzed.

Urinary metabolic profiles were acquired using ¹H NMR spectroscopy according to *Lau et al., 2018*. In brief one-dimensional 600 MHz ¹H NMR spectra of urine samples from each cohort were acquired on the same Bruker Avance III spectrometer operating at 14.1 Tesla within a period of 1 month. The spectrometer was equipped with a Bruker SampleJet system, and a 5 mm broad-band inverse configuration probe maintained at 300 K. Prior to analysis, cohort samples were randomized. Deuterated 3-(trimethylsilyl)-[2,2,3,3-d4]-propionic acid sodium salt (TSP) was used as an internal reference. Aliquots of the study pooled quality control (QC) sample were used to monitor analytical performance throughout the run and were analyzed at an interval of every 23 samples (i.e. four QC samples per well plate). The ¹H NMR spectra were acquired using a standard one-dimensional solvent suppression pulse sequence. 44 metabolites were identified and quantified as described (*Lau et al., 2018*). The urinary NMR showed excellent analytical performance, the mean coefficient of variation across the 44 NMR-detected urinary metabolites was 11%. Data were normalized using the median fold change normalization method (*Dieterle et al., 2006*), which takes into account the distribution of relative levels of all 44 metabolites compared to the reference sample in determining the most probable dilution factor. An offset of ½ of the minimal value was applied and then concentration levels were expressed as log2.

## Building biological clocks

Child epigenetic age was calculated based on Horvath's Skin and Blood clock (*Horvath et al., 2018*) using the methylclock R package (*Pelegí-Sisó et al., 2021*).

New transcriptome and immunometabolic clocks were trained against chronological age on transcriptome data and concatenated proteomic and metabolomic data respectively, from the HELIX subcohort children through elastic net regression, using the *glmnet* R package (*Zou and Hastie, 2005*). All 'omic data was first mean-centred and univariate scaled. To tune hyperparameters alpha and lambda, we performed a line search for alpha between 0 and 1, in 0.1 increments, and each time found the optimal value of lambda based on minimization of cross-validated mean squared error, using the *cvfits* function and 10-fold cross-validation. The best-performing combination of alpha and lambda was reserved for fitting the final model.

Transcriptome data and concatenated proteomic and metabolomic data from the HELIX panel study children were reserved for testing performance (Pearson's r and mean absolute error with chronological age) of the derived clocks. Paired, one-tailed t-tests were used to test if biological age measures increased between the HELIX subcohort and subsequent HELIX panel clinical examinations.

## Developmental measurements

During the HELIX subcohort examination, height and weight were measured using regularly calibrated instruments and converted to BMI and height age-and-sex–standardized z-scores (zBMI and zHeight) using the international World Health Organization (WHO) reference curves (*de Onis et al., 2007*). Bioelectric impedance analyses were performed with the Bodystat 1500 (Bodystat Ltd.) equipment after 5 min of lying down. The proportion of fat mass was calculated using published age- and race-specific equations validated for use in children (*Clasey et al., 2011*).

Trained fieldwork technicians measured three cognitive domains in children using a battery of computer-based tests: fluid intelligence (Raven Coloured Progressive Matrices Test [CPM]), attention

function (Attention Network Test [ANT]), and working memory (N-Back task). Complete outcome descriptions are provided in *Julvez et al., 2021*. The CPM comprised a total of 36 items and we used the total number of correct responses as the outcome. A higher CPM scoring indicates better fluid intelligence. Fluid intelligence is the ability to solve novel reasoning problems and depends only minimally on prior learning. For ANT, we used the outcome of hit reaction time standard error (HRT-SE), a measure of response speed consistency throughout the test. A high HRT-SE indicates highly variable reaction time during the attention task and is considered a measure of inattentiveness (*Forns et al., 2014*). As the main parameter of N-Back, we used d prime (d′) from the three-back colors test, a measure derived from signal detection theory calculated by subtracting the z-score of the false alarm rate from the z-score of the hit rate. A higher d′ indicates more accurate test performance, i.e., better working memory (*Forns et al., 2014*). All examiners were previously trained following a standardized assessment protocol by the study expert psychologist. Furthermore, during the pilot phase, a coordinator visited each cohort site and checked for any potential error committed by the previously trained examiners.

Parents completed questionnaires related to child's behavior, including the Conner rating scale (N=1287) and child behavior checklist (CBCL, N=1298), within a week before the follow-up visit at 6–11 years of age. The 99-item CBCL/6–18 version for school children was used to obtain standardized parent reports of children's problem behaviors, translated and validated in each native language of the participating six cohort populations (*Achenbach, 2001*). The parents responded along a three-point scale with the code of 0 if the item is not true of the child, one for sometimes true, and two for often true. The internalizing score includes the subscales of emotionally reactive and anxious/depressed symptoms, as well as somatic complaints and symptoms of being withdrawn. The externalizing score includes attention problems and aggressive behaviors.

Lung function was measured by a spirometry test (EasyOne spirometer; NDD [New Diagnostic Design], Zurich, Switzerland), by trained research technicians using a standardized protocol. The child, sitting straight and equipped with a nose clip, was asked to perform at least six maneuvres (if possible). Details of the exclusion of unacceptable maneuvers and validation of acceptable spirometer curves is fully described in *Agier et al., 2019*. FEV1 percent predicted values were computed using the reference equations estimated by the Global Lung Initiative (*Quanjer et al., 2012*), standardized by age, height, sex, and ethnicity.

Parents of children aged 8 years or older completed an additional questionnaire based on the pubertal development scale (PDS) (*Petersen et al., 1988*). Boys were asked whether growth has not begun, barely begun, is definitely underway, or has finished on five dimensions: body hair, facial hair, voice change, skin change, and a growth spurt. Girls were asked the same questions about body hair, skin change, breast development, and growth spurt. Responses were coded on four-point scales (1=no development and 4=completed development). For girls, a yes-no question about the onset of menarche is weighted more heavily (1=no and 4=yes). For both genders, ratings are then averaged to create an overall score for physical maturation. Due to the young age of participants, we took the average scores and created a binary variable, to define whether puberty had started (PDS >1) or not (PDS=1).

## Covariates

During pregnancy and in the childhood HELIX subcohort examination information on the following key covariates was collected: cohort study centre (BiB, EDEN, INMA, MoBa, KANC, and RHEA), self-reported maternal education (primary school, secondary school, and a university degree or higher), self-reported ancestry (White European, Asian and Pakistani, or other), birth weight (continuous, kg), gestational age at delivery (continuous in weeks).

Information about the children's habitual diet was collected via a semi-quantitative food-frequency questionnaire (FFQ) covering the child's habitual diet, which was filled in by the parent attending the examination appointment. The FFQ, covering the past year, was developed by the HELIX research group, translated, and applied to all cohorts. For the Mediterranean Diet Quality Index (KIDMED index) (*Serra-Majem et al., 2004*), items positively associated with the Mediterranean diet pattern (11 items) were assigned a value of +1, while those negatively associated with the Mediterranean diet pattern (four items) were assigned a value of −1. The scores for all 15 items were summed, resulting in a total KIDMED score ranging from −4–11, with higher scores reflecting greater adherence to a Mediterranean diet.

The smoking status of the mother at any point during pregnancy was categorized into 'non-active smoker,' or 'active smoker.' Global exposure of the child to environmental tobacco smoke was defined based on the questionnaires completed by the parents into: 'no exposure,' no exposure at home neither in other places; 'exposure:' exposure in at least one place, at home or outside. Moderate-to-vigorous physical activity variable was created based on the physical activity questionnaire developed by the HELIX research group. It was defined as the amount of time children spent doing physical activities with intensity above three metabolic equivalent tasks (METs) and is expressed in units of min/day.

Family Affluence Score (FAS) (*Boyce et al., 2006*) was included based on questions from the subcohort questionnaire. A composite FAS score was calculated based on the responses to the next four items: (1) Does your family own a car, van, or truck? (2) Do you have your own bedroom for yourself? (3) During the past 12 months, how many times did you travel away on holiday with your family? (4) How many computers does your family own? A three-point ordinal scale was used, where FAS low (score 0, 1, 2) indicates low affluence, FAS medium (score 3, 4, 5) indicates middle affluence, and FAS high (score 6, 7, 8, 9) indicates high affluence FAS.

Family social capital-related questions were included in the HELIX questionnaire to capture different aspects of social capital, relating both to the cognitive (feelings about relationships) and structural (number of friends, number of organizations) dimensions and to bonding capital (close friends and family), bridging capital (neighborhood connections, looser ties) and linking capital (ties across power levels; for example, political membership). Family social capital was categorized into low, medium, and high based on terciles.

## Statistical analysis

All statistical analyses described here were performed among the HELIX subcohort children only. Since there were few missing covariate data (*Supplementary file 2*), a complete-case analysis was performed. Correlations between biological age measures and chronological age were calculated using Pearson's correlations. Partial correlations, adjusted for chronological age and cohort study centre, were applied to assess correlations between biological age measures.

In analysis with health risk factors and developmental outcomes, relative telomere length was multiplied by –1 to provide directions of effect consistent with the biological age clocks and univariate scaled to express effects in terms of SD change in telomere length. The markers derived from omic-based biological clocks were expressed as Δ age (clock-predicted age – chronological age). Associations between the biological age markers and developmental measures were estimated using linear regression, or logistic regression for the onset of puberty, with the developmental measure as the dependent variable. CBCL scores were log-transformed to achieve an approximately normal distribution. Continuous outcomes, apart from the BMI and height z-scores, were mean-centered and univariate scaled for the purposes of graphical representation. Associations between health risk factors and biological age markers were estimated using linear regression with the biological age marker as the dependent variable. All regression analyses were adjusted for chronological age, sex, ethnicity, and study centre.

We preformed four sensitivity analyses: Firstly, we repeated the analysis with health outcomes stratified by child sex, since the relationship between biological age and development may differ between boys and girls. Secondly, we further adjusted regression models for estimated cell counts (CD4T, CD8T, monocytes, B cells, NK cells, neutrophils, and eosinophils), since it has been proposed for epigenetic clocks that cell proportion adjustments allow estimation of effects on the intrinsic cellular aging rate, rather than the extrinsic rate outputted by blood-based biological clocks, which may be partly determined by age-related changes in cell composition (*Chen et al., 2016*). Blood cell type proportion was estimated from DNA methylation data using the *He et al., 2014* reference panel as implemented in meffil package (*Costa et al., 2015*). Third, we assessed the effects of further adjustment for health risk factors identified as associated with any of the biological age markers (family affluence and social capital, birthweight, maternal active smoking, and child passive smoking). In our main analysis, we have not adjusted for these factors as our assumption is that the effects of health risk factors on child development are mediated through biological age. However, an alternative assumption is that health risk factors exert independent effects on both biological age and developmental outcomes, which would require adjustment for these factors to estimate the direct effects of biological age on developmental outcomes. Finally, we stratified by study centre to

check the consistency of effects among observed associations with developmental outcomes across study centres.

Due to the exploratory nature of the analysis, we report and discuss associations significant at both the nominal significance threshold (unadjusted p<0.05) and after correction for 5% false discovery rate using the *Benjamini and Hochberg, 1995* method, calculated across all computed associations (i.e. multiple testing corrected p-value <0.05).

We performed overrepresentation analyses (ORA) among KEGG and REACTOME pathways and gene ontology (GO) sets of all transcripts contributing to the transcriptome clock using the Consensus-pathDB online tool (http://consensuspathdb.org/). A pathway or GO set was considered significantly enriched if FDR-corrected p-values were smaller than 0.05 and included at least 3 genes. Additionally, to assess concordance with gene expression changes with age in adults, we tested the enrichment of all transcripts contributing to the transcriptome clock among age-associated transcripts reported by *Peters et al., 2015*, using a hypergeometric test using the R 'phyper' function.

All analyses were performed in R version 4.1.2.

## Results

### Sample characteristics

We used blood or urine-derived measurements from the pan-European HELIX cohort. This included blood telomere length (N=1162), blood DNA methylation (N=1173, 450 K CpGs), blood gene expression (N=1007, 50 K genes), and proteins and metabolites (N=1152, 36 plasma proteins, 177 serum metabolites, and 44 urinary metabolites), with 869 children overlapping across all measurements. Each subsample included around 55% boys, 89% children of white European ancestry, and a mean age of around 8 years (range 5–12 years). Around 51% of mothers of the HELIX children in each subsample had a high education level. The HELIX cohort included children from six study centres based in the UK, Spain, Greece, Lithuania, France, and Norway, with each centre contributing between 11 and 24% to each subsample (see *Table 1* for sample characteristics).

### Biological age marker performance

We included two established markers of biological age, telomere length, DNA methylation age, and developed two new candidate biological age markers, transcriptome age and 'immunometabolic' age (*Figure 2*). DNA methylation age was calculated using the published Skin and blood Horvath clock (*Horvath et al., 2018*) to allow greater comparison to the wider literature, including in adults. We previously reported this epigenetic clock to show the best performance in chronological age prediction within the HELIX cohort (*de Prado-Bert et al., 2021*). Since no published applicable transcriptome, proteome or metabolome clocks were available for the age range of our sample, we trained two new biological clocks using these data in the HELIX cohort, through elastic net regression and cross-validation. We combined the proteome and metabolome data into a single immunometabolic age clock, since the available proteomic data included biomarkers targeting both metabolic and inflammatory functions, both omic types represent final products of gene regulation, and since the metabolic and immune systems are closely linked (*Hotamisligil, 2017*).

The correlation between telomere length and chronological age was weak but statistically significant (r=–0.07, p=0.02). Correlations with chronological age were r=0.85 for DNA methylation age, r=0.94 for transcriptome age, and r=0.86 for immunometabolic age (*Figure 2*) .

We validated the transcriptome and immunometabolic clocks using cross-validation within the HELIX subcohort (cross-validated r of 0.87 and 0.82, respectively) and further tested in a subset of children who attended a second clinic visit approximately 0.5 years after the main follow-up visit (standard deviation (SD)=0.18 years) as part of the HELIX panel study. Correlations in this test set were r=0.93 for transcriptome age (N=128) and r=0.84 for immunometabolic age (N=151) (*Figure 2*). Predicted biological age increased by mean of 0.33 years (SD=0.58) for transcriptome age (t-test, p=3 e–5) and a mean of 0.22 years (SD: 0.59 years) for immunometabolic age (t-test, p=2 e–5) between the first and second visits (*Figure 2—figure supplement 1*). Correlations were significant (p<0.05) within each study centre for both clocks, except for immunometabolic age for children from the BiB (UK) cohort (*Figure 2—figure supplement 2* and *Figure 2—figure supplement 3*).

**Table 1.** Summary Statistics for the study population.

| | Telomere Length | DNA methylation age | Trancript-ome age | Immuno-metabolic age |
|---|---|---|---|---|
| | N (%) or Mean (SD) | N (%) or Mean (SD) | N (%) or Mean (SD) | N (%) or Mean (SD) |
| N | 1162 | 1173 | 1007 | 1152 |
| Demographic factors | | | | |
| Age (years) | 7.84 (1.54) | 7.84 (1.54) | 7.90 (1.50) | 7.86 (1.55) |
| Sex-Male | 639 (55) | 644 (54.9) | 547 (54.3) | 628 (54.5) |
| Sex-Female | 523 (45) | 529 (45.1) | 460 (45.7) | 524 (45.5) |
| Ethnicity-White | 1039 (89.4) | 1048 (89.3) | 905 (89.9) | 1032 (89.6) |
| Ethnicity-Pakistani/Asian | 96 (8.3) | 98 (8.4) | 76 (7.5) | 93 (8.1) |
| Ethnicity -Other | 27 (2.3) | 27 (2.3) | 26 (2.6) | 27 (2.3) |
| Cohort-BIB | 200 (17.2) | 203 (17.3) | 162 (16.1) | 191 (16.6) |
| Cohort-EDEN | 145 (12.5) | 146 (12.4) | 109 (10.8) | 149 (12.9) |
| Cohort-INMA | 212 (18.2) | 215 (18.3) | 184 (18.3) | 201 (17.4) |
| Cohort-KANC | 196 (16.9) | 198 (16.9) | 151 (15) | 197 (17.1) |
| Cohort-MOBA | 211 (18.2) | 212 (18.1) | 245 (24.3) | 222 (19.3) |
| Cohort-RHEA | 198 (17) | 199 (17) | 156 (15.5) | 192 (16.7) |
| Prenatal factors | | | | |
| maternal non-active smoker during pregnancy | 988 (85) | 998 (85.1) | 859 (85.3) | 981 (85.2) |
| Maternal active smoker during pregnancy | 174 (15) | 175 (14.9) | 148 (14.7) | 171 (14.8) |
| Birthweight (kg) | 3.37 (0.5) | 3.37 (0.5) | 3.38 (0.52) | 3.38 (0.5) |
| Gestational age (weeks) | 39.57 (1.67) | 39.58 (1.67) | 39.59 (1.75) | 39.59 (1.66) |
| Family Capital | | | | |
| Maternal Education (low) | 165 (14.7) | 166 (14.7) | 140 (14.4) | 157 (14.1) |
| Maternal Education (medium) | 391 (34.8) | 394 (34.8) | 328 (33.8) | 391 (35.1) |
| Maternal Education (high) | 568 (50.5) | 573 (50.6) | 503 (51.8) | 565 (50.8) |
| Family Affluence (low) | 133 (11.5) | 135 (11.5) | 112 (11.1) | 128 (11.1) |
| Family Affluence (medium) | 462 (39.8) | 466 (39.8) | 394 (39.2) | 450 (39.1) |
| Family Affluence (high) | 565 (48.7) | 570 (48.7) | 499 (49.7) | 572 (49.7) |
| Family Social Capital (low) | 513 (47.7) | 516 (47.5) | 422 (45.8) | 496 (46.7) |
| Family Social Capital (medium) | 264 (24.6) | 269 (24.8) | 228 (24.7) | 259 (24.4) |
| Family Social Capital (high) | 298 (27.7) | 301 (27.7) | 272 (29.5) | 307 (28.9) |
| Child factors | | | | |

*Table 1 continued on next page*

*Table 1 continued*

| | Telomere Length | DNA methylation age | Trancript-ome age | Immuno-metabolic age |
|---|---|---|---|---|
| No passive smoke exposure | 723 (63.8) | 732 (63.9) | 639 (64.5) | 718 (63.8) |
| Passive smoke exposure | 411 (36.2) | 413 (36.1) | 351 (35.5) | 407 (36.2) |
| Physical Activity-Low | 418 (36.9) | 420 (36.8) | 349 (35.3) | 416 (37.1) |
| Physical Activity-Medium | 336 (29.7) | 341 (29.9) | 295 (29.9) | 330 (29.4) |
| Physical Activity-High | 378 (33.4) | 381 (33.4) | 344 (34.8) | 375 (33.5) |
| KIDMED diet score | 2.81 (1.77) | 2.82 (1.78) | 2.88 (1.77) | 2.84 (1.76) |
| Developmental measures | | | | |
| Height z-score | 0.4 (0.97) | 0.39 (0.98) | 0.39 (0.96) | 0.4 (0.98) |
| BMI z-score | 0.43 (1.2) | 0.43 (1.2) | 0.4 (1.15) | 0.42 (1.18) |
| Adiposity (BIA fat-mass %) | 6.76 (4.01) | 6.77 (4.01) | 6.52 (3.9) | 6.72 (3.95) |
| Working memory (3-back d') | 1.1 (1.01) | 1.1 (1.01) | 1.13 (1) | 1.1 (1.01) |
| Inattentiveness (ANT-HRT) | 301.97 (90.38) | 301.93 (90.46) | 297.69 (89.36) | 301.35 (89.84) |
| Fluid Intelligence (CPM) | 25.87 (6.33) | 25.86 (6.32) | 26.12 (6.26) | 25.95 (6.3) |
| Internalizing behaviors (CBCL) | 6.49 (5.9) | 6.48 (5.9) | 6.36 (5.89) | 6.52 (5.87) |
| Externalizing behaviors (CBCL) | 6.81 (6.5) | 6.82 (6.51) | 6.67 (6.49) | 6.74 (6.42) |
| Lung Function (FEV1) | 99.26 (13.46) | 99.25 (13.47) | 99.16 (13.02) | 99.17 (13.47) |
| Puberty not started | 250 (46.6) | 252 (46.5) | 254 (49.7) | 260 (48) |
| Puberty started (PDS >1) | 287 (53.4) | 290 (53.5) | 257 (50.3) | 282 (52) |

The immunometabolic age clock was composed of 135 predictors including 20 proteins, 79 serum metabolites, and 36 urinary metabolites (*Supplementary file 3*). The transcriptome clock was composed of 1445 genes, 652 of which were annotated to Gene Symbols ( *Supplementary file 4*). The transcriptome clock genes were enriched (false discovery rate (FDR)-corrected p<0.05) in 'ribosome' and 'ribosome biogenesis' KEGG pathways (*Supplementary file 5*) and the following level 2 Gene Ontology biological process terms: 'leukocyte activation,' 'movement of cell or subcellular component,' 'leukocyte migration,' 'cell activation,' and 'secretion by cell' (*Supplementary file 6*). We also tested the enrichment of transcriptome clock predictors among genes reported by a large meta-analysis of age in adults (*Peters et al., 2015*): among the 1406 reported age-associated genes that could be matched to our measured genes, 43 were included in our transcriptome clock (hypergeometric enrichment test, p=0.052). We note that since a common definition of markers of biological age is that they should be associated with age-related disease and mortality (*Moskalev, 2019*) these new clocks may only currently be considered 'candidate' biological age markers. However, we have referred to both the established and candidate markers as biological age markers throughout to simplify the presentation.

*Figure 3* shows partial correlations, adjusted for chronological age and study centre, between the biological age markers. Only null to weak correlations were observed, with significant correlations between telomere length and DNA methylation age ($r=-0.06$, $p=0.04$) and between transcriptome age and immunometabolic age ($r=0.08$, $p=0.01$).

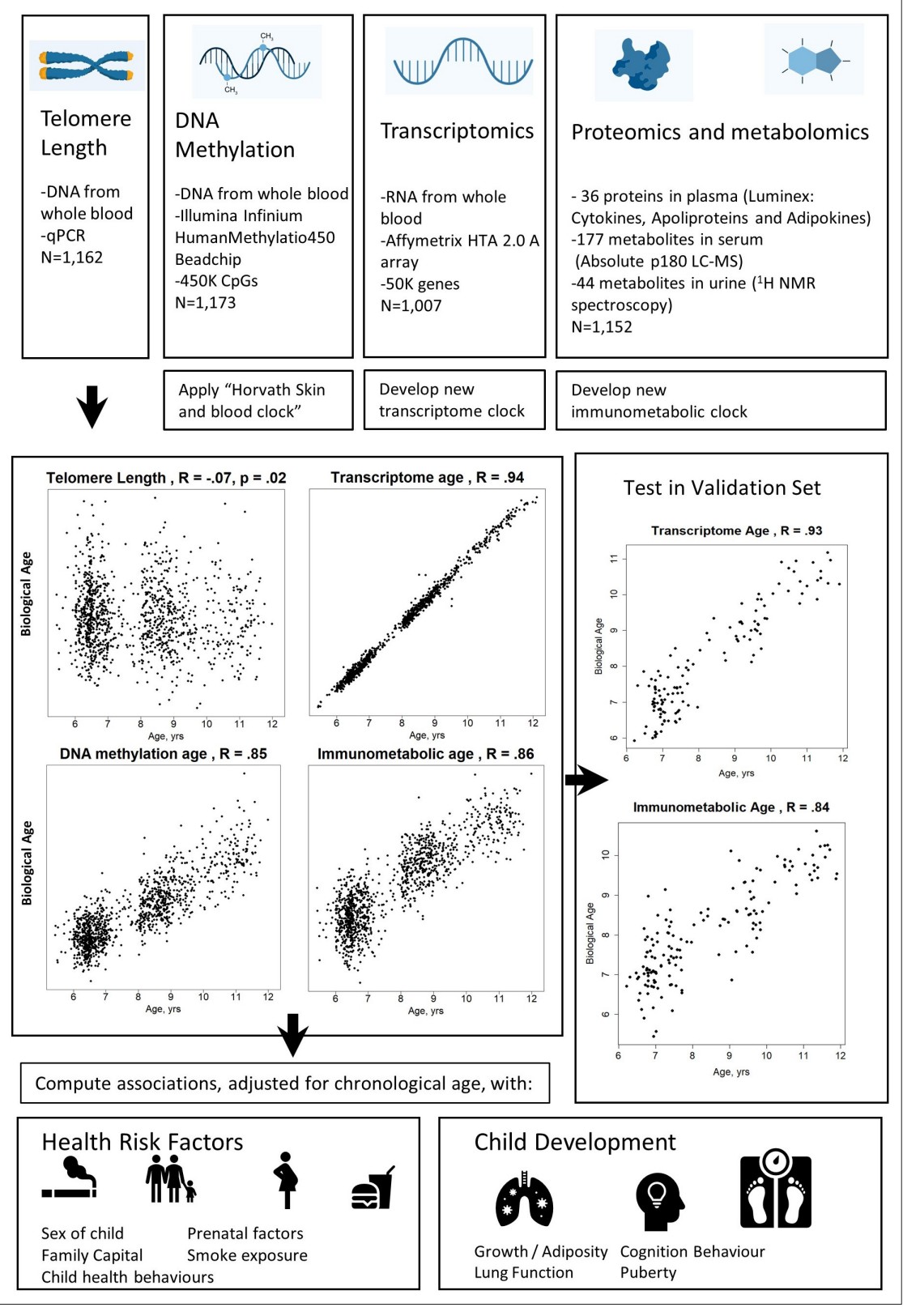

**Figure 2.** Study design schematic. Source data for reproducing correlation plots are provided in *Figure 2—source data 1*.

The online version of this article includes the following source data and figure supplement(s) for figure 2:

**Source data 1.** Source data for reproducing correlation plots in *Figure 2*.

*Figure 2 continued on next page*

*Figure 2 continued*

**Figure supplement 1.** Comparison between immunometabolic and transcriptome age between first and second study visits.

**Figure supplement 2.** Age Prediction by study centre of transcriptome age.

**Figure supplement 3.** Age Prediction by study centre of immunometabolic age.

### Biological clock associations with health risk factors

*Table 2* shows associations, adjusted for chronological age, sex, study centre and ethnicity, between health risk factors and the biological age markers. The markers derived from omic-based biological clocks are expressed as Δ age (clock-predicted age – chronological age) and since the adjustment set included chronological age, effects can be interpreted as years of age acceleration as often defined (*Jansen et al., 2021*).

Nominally significant associations were observed for the following health risk factors: Telomere length was longer among girls compared to boys (p**=**3 e–06) and among children of high affluence families (p=0.008). DNA methylation Δ age was higher among children of mothers who actively smoked during pregnancy (p=0.018) and children exposed to passive smoke (p=0.023), while

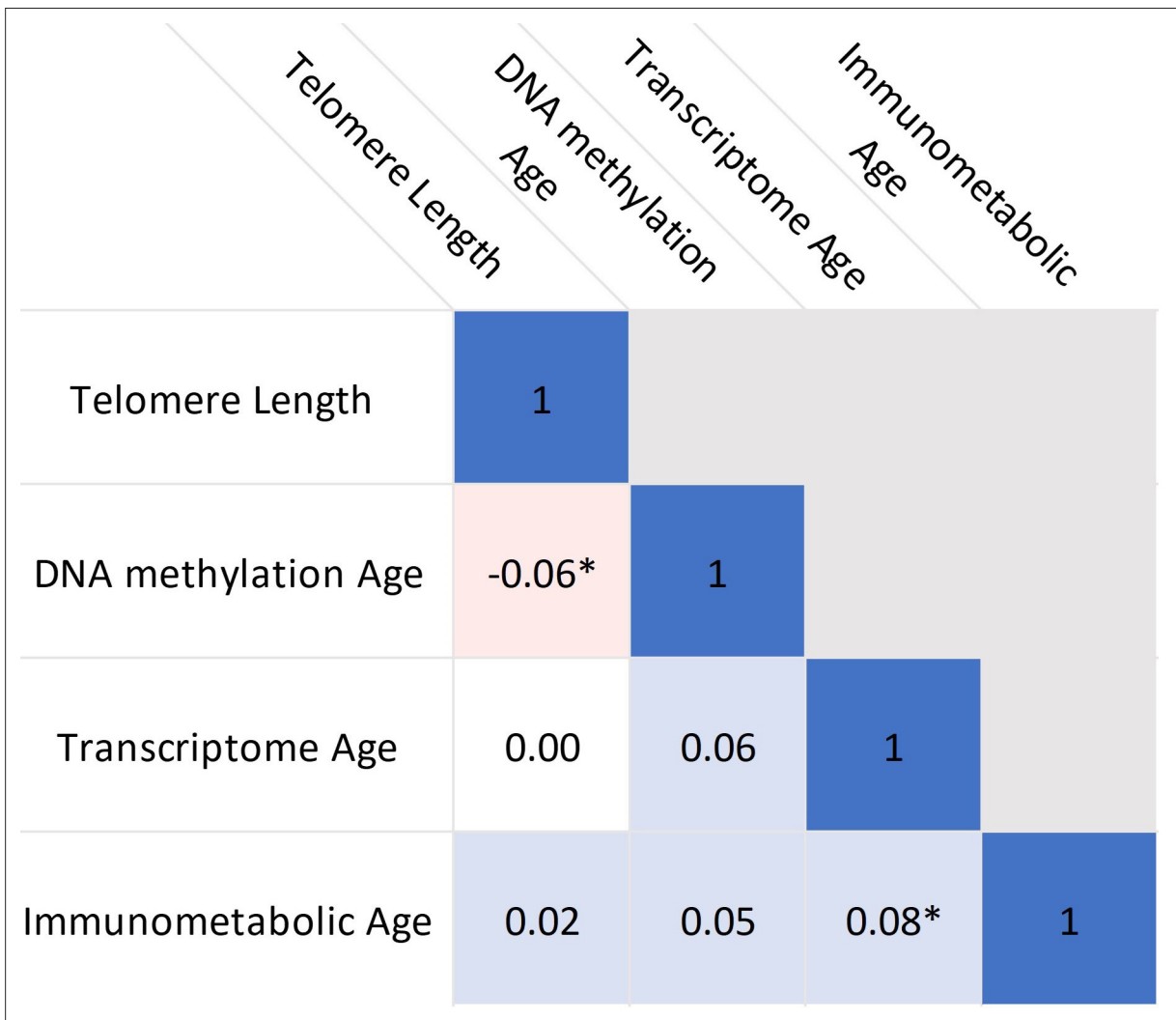

**Figure 3.** Correlations between biological age indicators. Heatmap shows partial Pearson's correlations, adjusted for chronological age and study centre. * indicates p<0.05. Source data for reproducing plots is provided in *Figure 3—source data 1*.

The online version of this article includes the following source data for figure 3:

**Source data 1.** Source data for reproducing *Figure 3*.

**Table 2.** Associations between health risk factors and biological age measures.

Estimates were calculated using linear regression, adjusted for chronological age, sex, ethnicity, and study centre. Bold indicates p<0.05 and *indicates FDR <5%. Telomere length is expressed as a standard deviation (SD) decrease in length (multiplied by −1) to provide estimates indicative of accelerated biological age, as the other biological age indicators. Telomere Length N=1162, DNA methylation age N=1173, Transcriptome age N=1007, Immunometabolic age N=1152.

| | Telomere Length | | DNA methylation age | | TranScriptome age | | Immunometabolic age | |
|---|---|---|---|---|---|---|---|---|
| | SD Decrease (95% CI) | p-value | Increase in years Δ Age (95% CI) | p-value | Increase in years Δ Age (95% CI) | p-value | Increase in years Δ Age (95% CI) | p-value |
| Sex-Male | - | - | - | - | - | - | - | - |
| Sex-Female | −0.27 (−0.39,−0.16) | 3.30E-06* | 0.07 (-0.01, 0.16) | 0.1 | 0 (-0.01, 0.02) | 0.73 | 0.06 (-0.01, 0.13) | 0.086 |
| Prenatal factors | | | | | | | | |
| maternal non-active smoker during pregnancy | - | - | - | - | - | - | - | - |
| Maternal active smoker during pregnancy | 0.07 (-0.1, 0.23) | 0.41 | 0.15 (0.03, 0.28) | **0.018** | 0 (-0.02, 0.02) | 0.88 | −0.04 (-0.14, 0.06) | 0.43 |
| Birthweight (kg) | −0.098 (-0.218, 0.023) | 0.11 | −0.021 (-0.114, 0.072) | 0.66 | 0.005 (-0.01, 0.02) | 0.51 | 0.102 (0.027, 0.177) | **0.0075** |
| Gestational age (weeks) | −0.012 (-0.048, 0.024) | 0.52 | 0.013 (-0.015, 0.041) | 0.35 | 0 (-0.005, 0.004) | 0.89 | 0.018 (-0.005, 0.04) | 0.12 |
| Family Capital | | | | | | | | |
| Maternal Education (low) | - | - | - | - | - | - | - | - |
| Maternal Education (medium) | −0.06 (-0.26, 0.13) | 0.53 | 0.02 (-0.14, 0.17) | 0.84 | 0.01 (-0.02, 0.03) | 0.61 | 0.08 (-0.04, 0.2) | 0.21 |
| Maternal Education (high) | −0.1 (-0.29, 0.1) | 0.32 | −0.07 (-0.22, 0.08) | 0.37 | 0 (-0.02, 0.03) | 0.85 | 0.12 (0, 0.24) | 0.051 |
| Family Affluence (low) | - | - | - | - | - | - | - | - |
| Family Affluence (medium) | −0.15 (-0.34, 0.05) | 0.13 | −0.11 (-0.26, 0.03) | 0.13 | 0 (-0.03, 0.02) | 0.85 | 0.02 (-0.1, 0.14) | 0.8 |

*Table 2 continued on next page*

*Table 2 continued*

| | Telomere Length | | DNA methylation age | | TranScriptome age | | Immunometabolic age | |
|---|---|---|---|---|---|---|---|---|
| Family Affluence (high) | −0.27 (−0.47, −0.07) | **0.0081** | −0.14 (−0.29, 0.02) | 0.083 | 0.01 (−0.01, 0.04) | 0.35 | 0.09 (−0.04, 0.21) | 0.17 |
| Family Social Capital (low) | - | | - | | - | | - | |
| Family Social Capital (medium) | −0.06 (−0.21, 0.09) | 0.45 | −0.03 (−0.14, 0.09) | 0.62 | 0.02 (0.01, 0.04) | **0.012** | −0.04 (−0.14, 0.05) | 0.36 |
| Family Social Capital (high) | −0.15 (−0.3, 0) | 0.054 | −0.12 (−0.23, 0) | **0.048** | 0.02 (0.01, 0.04) | **0.011** | −0.06 (−0.15, 0.04) | 0.25 |
| Child factors | | | | | | | | |
| No passive smoke exposure | - | | - | | - | | - | |
| Passive smoke exposure | 0.05 (−0.08, 0.18) | 0.42 | 0.11 (0.02, 0.21) | **0.023** | 0.01 (0, 0.03) | 0.16 | −0.01 (−0.09, 0.07) | 0.76 |
| Physical Activity-Low | - | | - | | - | | - | |
| Physical Activity-Medium | 0.09 (−0.06, 0.23) | 0.25 | −0.08 (−0.2, 0.03) | 0.15 | −0.01 (−0.03, 0.01) | 0.17 | 0.03 (−0.06, 0.12) | 0.56 |
| Physical Activity-High | 0.14 (−0.01, 0.29) | 0.067 | −0.1 (−0.22, 0.01) | 0.08 | 0 (−0.02, 0.01) | 0.69 | −0.06 (−0.15, 0.04) | 0.24 |
| KIDMED diet score | −0.03 (−0.064, 0.005) | 0.092 | 0.005 (−0.022, 0.031) | 0.74 | 0.004 (−0.001, 0.008) | 0.10 | −0.005 (−0.027, 0.016) | 0.64 |

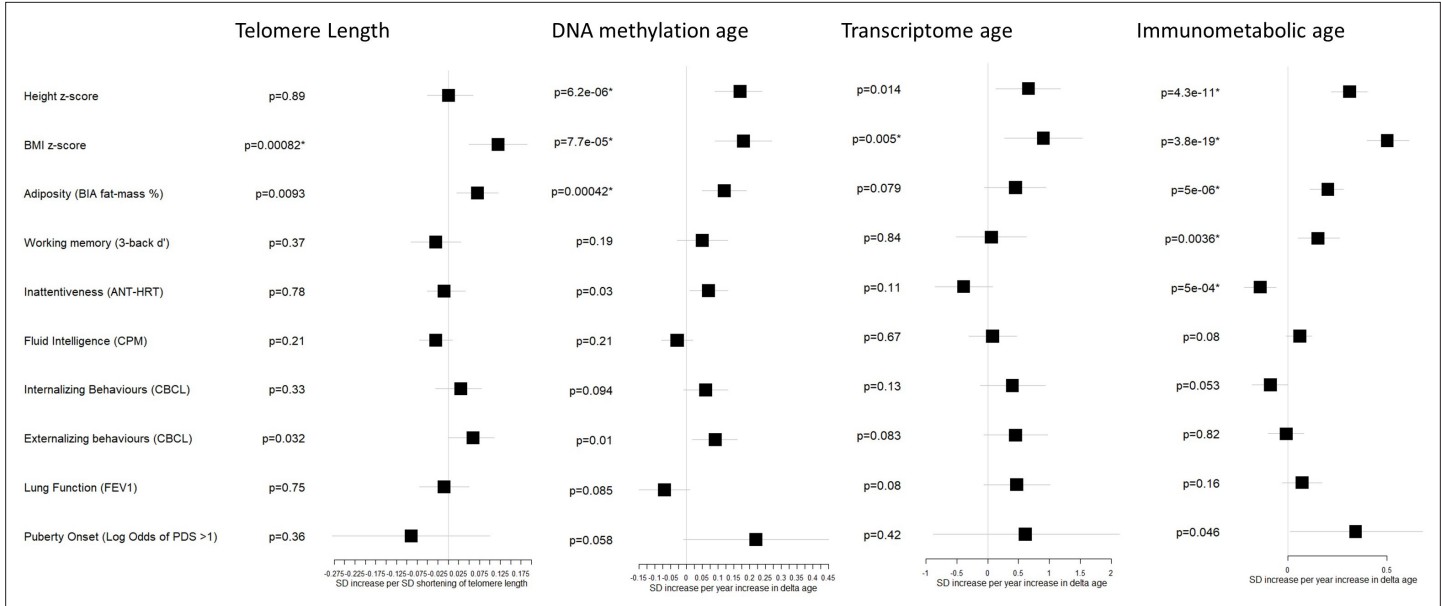

**Figure 4.** Associations between biological age measures and developmental measures. Estimates were calculated using linear regression, adjusted for chronological age, sex, ethnicity, and study centre. *indicates FDR <5%. Telomere length is expressed as a standard deviation (SD) decrease in length (multiplied by –1) to provide estimates indicative of accelerated biological age, as the other biological age indicators. Error bars show 95% confidence intervals. See *Table 3* for numbers included in each analysis and exact point estimates and confidence intervals.

The online version of this article includes the following figure supplement(s) for figure 4:

**Figure supplement 1.** Associations between biological age measures and developmental measures, stratified by sex.

**Figure supplement 2.** Associations between telomere length and developmental measures adjusted for (**A**) chronological age, sex, ethnicity, and study centre; (**B**) as for A plus estimated cell counts; (**C**) as for A plus family affluence and social capital, birthweight, maternal active smoking, and child passive smoking; (**D**) as for C plus estimated cell counts.

**Figure supplement 3.** Associations between DNA methylation Δ age and developmental measures adjusted for (**A**) chronological age, sex, ethnicity, and study centre; (**B**) as for A plus estimated cell counts; (**C**) as for A plus family affluence and social capital, birthweight, maternal active smoking, and child passive smoking; (**D**) as for C plus estimated cell counts.

**Figure supplement 4.** Associations between transcriptome Δ age and developmental measures adjusted for (**A**) chronological age, sex, ethnicity, and study centre; (**B**) as for A plus estimated cell counts; (**C**) as for A plus family affluence and social capital, birthweight, maternal active smoking, and child passive smoking; (**D**) as for C plus estimated cell counts.

**Figure supplement 5.** Associations between immunometabolic Δ age and developmental measures adjusted for (**A**) chronological age, sex, ethnicity, and study centre; (**B**) as for A plus estimated cell counts; (**C**) as for A plus family affluence and social capital, birthweight, maternal active smoking, and child passive smoking; (**D**) as for C plus estimated cell counts.

**Figure supplement 6.** Associations between biological age measures and developmental measures, stratified by study centre (adjusted for chronological age, sex, and ethnicity).

DNA methylation Δ age was lower among children from families with high social capital (p=0.048). Conversely, transcriptome Δ age was positively associated with medium and high (p=0.011) family social capital. Immunometabolic Δ age was associated with higher birthweight (p=0.0075). Only the association between longer telomere length and female sex passed the FDR correction.

## Biological age associations with development

*Figure 4* and *Table 3* shows associations, adjusted for chronological age, sex, study centre and ethnicity, between the biological age markers and developmental outcomes related to growth and adiposity, cognition, behavior, lung function, and the onset of puberty.

Several developmental outcomes were associated with biological age markers after FDR correction: DNA methylation and immunometabolic Δ age were associated with greater height z-score (p=6 e–6 and p=4 e–11, respectively) and greater fat mass % (p=0.0004 and p=5 e–6, respectively). All biological age markers were associated with greater BMI z-score (telomere length p=8 e–4, DNA methylation Δ age p=8 e–5, transcriptome Δ age p=0.005, immunometabolic Δ age p=4 e–19). Furthermore,

**Table 3.** Associations between biological age measures and developmental measures. Estimates were calculated using linear regression, adjusted for chronological age, sex, ethnicity, and study centre.

| | Telomere Length | | | DNA methylation age | | | Transcriptome age | | | Immunometabolic age | | |
|---|---|---|---|---|---|---|---|---|---|---|---|---|
| | N | SD increase / odds ratio * per SD shortening (95% CI) | p-value | N | SD increase / odds ratio per year increase in Δ age (95% CI) | p-value | N | SD increase / odds ratio per year increase in Δ age (95% CI) | p-value | N | SD increase/ odds ratio per year increase in Δ age (95% CI) | p-value |
| Height z-score | 1162 | 0 (-0.05, 0.06) | 0.89 | 1173 | 0.17 (0.09, 0.24) | 6.20E-06* | 1007 | 0.66 (0.13, 1.18) | 0.014 | 1152 | 0.31 (0.22, 0.4) | 4.30E-11* |
| BMI z-score | 1162 | 0.12 (0.05, 0.19) | 0.00082* | 1173 | 0.18 (0.09, 0.27) | 7.70E-05* | 1007 | 0.9 (0.27, 1.53) | 0.005* | 1152 | 0.5 (0.4, 0.61) | 3.80E-19* |
| Adiposity (BIA fat-mass %) | 1153 | 0.07 (0.02, 0.12) | 0.0093* | 1164 | 0.12 (0.05, 0.19) | 0.0004* | 999 | 0.45 (-0.05, 0.94) | 0.079 | 1144 | 0.2 (0.11, 0.28) | 5.00E-06* |
| Working memory (3-back d') † | 882 | -0.03 (-0.09, 0.03) | 0.37 | 890 | 0.05 (-0.03, 0.13) | 0.19 | 784 | 0.06 (-0.51, 0.63) | 0.84 | 876 | 0.15 (0.05, 0.26) | 0.0036* |
| Inattentiveness (ANT-HRT) | 1142 | -0.01 (-0.05, 0.04) | 0.78 | 1153 | 0.07 (0.01, 0.13) | 0.03 | 997 | -0.39 (-0.85, 0.08) | 0.11 | 1135 | -0.14 (-0.22, -0.06) | 5.00E-04* |
| Fluid Intelligence (CPM) | 1156 | -0.03 (-0.07, 0.01) | 0.21 | 1167 | -0.03 (-0.08, 0.02) | 0.21 | 1001 | 0.08 (-0.3, 0.47) | 0.67 | 1147 | 0.06 (-0.01, 0.12) | 0.08 |
| Internalizing Behaviors (CBCL) | 1156 | 0.03 (-0.03, 0.08) | 0.33 | 1166 | 0.06 (-0.01, 0.13) | 0.094 | 1002 | 0.4 (-0.12, 0.93) | 0.13 | 1146 | -0.09 (-0.18, 0) | 0.053 |
| Externalizing behaviors (CBCL) | 1156 | 0.06 (0, 0.11) | 0.032 | 1166 | 0.09 (0.02, 0.16) | 0.01 | 1002 | 0.45 (-0.06, 0.97) | 0.083 | 1146 | -0.01 (-0.1, 0.08) | 0.82 |
| Lung Function (FEV1) | 911 | -0.01 (-0.07, 0.05) | 0.75 | 921 | -0.07 (-0.15, 0.01) | 0.085 | 795 | 0.47 (-0.06, 1.01) | 0.08 | 907 | 0.07 (-0.03, 0.17) | 0.16 |
| Puberty onset ‡ | 537 | 0.92 (0.76, 1.11) | 0.36 | 542 | 1.25 (0.99, 1.57) | 0.058 | 511 | 1.84 (0.41, 8.44) | 0.42 | 542 | 1.41 (1.01, 1.97) | 0.046 |

Bold indicates p<0.05 and *indicates FDR <5%.

*Odds ratio provided for puberty onset only.

†Not available in the Lithuanian KANC cohort.

‡Only assessed in children over 8 years old.

immunometabolic Δ age was associated after FDR correction with better working memory (p=0.0036) and reduced inattentiveness (p=5 e–4).

Associations at the nominal significance (p<0.05) level was observed for increases in height z-score with transcriptome Δ age (p=0.014), shorter telomere length with increased fat mass % (p=0.009), and DNA methylation Δ age with greater inattentiveness (p=0.03). Both shorter telomere length and DNA methylation Δ age were associated with greater externalizing behaviors (p=0.032 and p=0.01, respectively). Among a smaller subset of children (*Table 1*) aged over 8 years, we observed a nominally significant association between immunometabolic Δ age and odds of onset of puberty (Odds Ratio: 1.41, 95% CI: 1.01, 1.97, p=0.046).

No significant associations with lung function were observed, but like the patterns of associations observed with cognitive and behavioral outcomes, there was a trend for a negative association with DNA methylation Δ age (p=0.085) and a positive association with immunometabolic Δ age (p=0.16).

## Sensitivity analysis

In sensitivity analysis. We first stratified by sex and generally observed similar associations among boys and girls, apart from the following differences (*Figure 4—figure supplement 1*): Associations between shorter telomere length and BMI z-score and adiposity were stronger among boys. For DNA methylation Δ age, associations with poorer externalizing and internalizing behaviours were only apparent among boys. For transcriptome Δ age, stronger associations among boys were observed with BMI z-score, adiposity, and poorer externalizing and internalizing behaviors. Conversely, we observed an association between transcriptome Δ age and reduced inattentiveness among girls only. Immunometabolic Δ age was more strongly associated with reduced inattentiveness among girls and also associated with greater odds of puberty onset among girls only.

Second, we additionally adjusted our models by estimating cell counts to determine the influence of cell composition on associations with developmental outcomes (*Figure 4—figure supplement 2B*, *Figure 4—figure supplement 3B*, *Figure 4—figure supplement 4B* and *Figure 4—figure supplement 5B*; *Supplementary file 7*). Associations were generally little changed: For DNA methylation Δ age, associations were attenuated with adiposity and growth outcomes although all remained FDR significant and the association with externalizing behavior was slightly attenuated. For transcriptome, Δ age associations with adiposity and growth outcomes and lung function increased slightly and the association with greater lung function became nominally significant.

Third, we assessed the effects of further adjustment for health risk factors (family affluence and social capital, birthweight, maternal active smoking, and child passive smoking) since health risk factors could be independently associated with both biological age and developmental outcomes (*Figure 4—figure supplement 2C*, *Figure 4—figure supplement 3C*, *Figure 4—figure supplement 4C* and *Figure 4—figure supplement 5C*; *Supplementary file 7*). Associations were generally little changed, expect for an attenuation of the association between telomere length and externalizing behavior, while conversely the association between DNA methylation Δ age and externalizing behavior was slightly strengthened.

Finally, we stratified by cohort study centre, observing generally consistent directions of effect among developmental outcomes of at least at nominal significance (p<0.05) in the main pooled analysis (*Figure 4—figure supplement 6*).

## Discussion

In a large sample of European children, we have analyzed two established and two candidate measures of biological age, derived from molecular features at different levels of biological organization, in relation to developmental outcomes and health risk factors. We assessed two established biological age markers, telomere length and DNA methylation age, and derived two new measures, transcriptome age, and immunometabolic age. Despite finding only null to weak correlations between the measures, we found all measures to be positively associated with greater BMI and adiposity, and both DNA methylation Δ age and immunometabolic Δ age were associated with taller height. While immunometabolic Δ age was associated with greater cognitive maturity including greater working memory and attentiveness, conversely DNA methylation Δ age was nominally associated with greater

inattentiveness and both DNA methylation Δ age and shorter telomere length were nominally associated with poorer externalizing behaviors.

BMI has consistently been associated with accelerated aging in adults across a diverse range of biological age markers (*Peters et al., 2015*; *Jansen et al., 2021*; *Nevalainen et al., 2017*; *Fiorito et al., 2019*) underlining the integral role of metabolism in aging. Indeed, a recent large study of Dutch adults found BMI to be the only health risk factor tested associated with accelerated aging across five biological age clocks, including telomere length, DNA methylation, transcriptome, proteome, and metabolomic age markers (*Jansen et al., 2021*). Here, we show that the link between BMI and multiple dimensions of accelerated aging is also apparent in children. Energy and nutrient intake influence all Ageing Hallmarks and multiple lines of evidence link increased adiposity to shorter lifespan (*López-Otín et al., 2016*). These effects appear to be partially mediated through evolutionarily conserved nutrient-sensing systems such as the mTOR signaling pathway, which promote anti-aging cellular repair mechanisms, at the expense of growth and metabolism, in response to lower nutrient availability (*López-Otín et al., 2016*). Furthermore, excess adiposity increases generalized inflammation and oxidative stress (*Suzuki et al., 2003*; *Minamino et al., 2009*), which may have direct effects on age markers, particularly telomere length and DNA methylation age acceleration.

The observed associations between greater height with biological age may indicate developmental maturity. Height is generally considered reflective of a beneficial early-life environment (*Davey Smith et al., 2000*), however, evidence for an association with lifespan is mixed (*Davey Smith et al., 2000*; *Tanisawa et al., 2018*), with a recent meta-analysis suggesting a u-shaped relationship with all-cause mortality (*Li et al., 2021*). Greater comparative height at 10 years was also inversely associated with longevity in a recent large-scale Medelian randomization study (*Huang et al., 2021*). Furthermore, there is some evidence that the link between height and longevity may be mediated through the insulin-like growth factor-1 signaling pathway (*Tanisawa et al., 2018*; *He et al., 2014*). The associations may also be interpreted as greater rates of growth and anabolism exerting greater 'wear and tear' on cellular structures. Two other studies have also observed an association between height and DNA methylation age acceleration in children (*Simpkin et al., 2017*; *Suarez et al., 2018*).

Despite similarities in associations with growth and adiposity measures, patterns of association across cognitive and behavioral domains varied across biological age markers, underlying the view of biological aging as a multi-faceted process. Immunometabolic Δ age was associated with greater cognitive maturity, fitting the life-course model of greater accumulation of biological capital during the build-up phase of development. Immunometabolic Δ age may be considered a phenotypic summary measure of metabolic and immune system maturity, and these cognitive developmental associations suggest that it may also be generalizable to the overall developmental stage. On the other hand, DNA methylation Δ age was related to relative immaturity in attentiveness and externalizing behavior. A previous Finish study of children aged between 11 and 13 years also reported associations between DNA methylation age acceleration and behavioral problems (*Suarez et al., 2018*). Similarly, shorter telomere length was associated here with greater externalizing behaviors, although not with any cognitive domains. Four other studies have examined the link between shorter telomere length and externalizing behaviors (*Costa et al., 2015*; *Daoust et al., 2023*; *Wojcicki et al., 2015*; *Kroenke et al., 2011*), with all, except one (*Kroenke et al., 2011*), also reporting an association.

Overall patterns of associations between risk factors and biological age measures also suggest the detrimental nature of accelerated aging in children assessed through telomere length and DNA methylation Δ age, and the potentially beneficial nature of advanced immunometabolic Δ age. Both prenatal maternal active smoking and child passive smoking were associated with DNA methylation Δ age, while greater birthweight was associated with immunometabolic Δ age. We examined maternal education level, family affluence, and social capital which broadly represent the three forms of inter-convertible capital (cultural, economic, and social) proposed by *Bourdieu, 1986*. It has been theorized that biological capital represents a fourth type of human capital, and that the conversion across these forms of capital underlies inequalities in aging trajectories (*Vineis and Kelly-Irving, 2019*). Nominally significant associations between higher family affluence with longer telomere length and high social capital with a younger DNA methylation age indicate that age acceleration assessed through these measures does not represent an accumulation of biological capital. Generally, directions of effect for immunometabolic Δ age were in the opposite direction which may suggest it represents greater biological capital.

Girls were found to have longer telomere lengths than boys. Women have been consistently found to have longer telomere lengths (*Gardner et al., 2014*) although the few generally smaller studies in children have been inconsistent (*Wojcicki et al., 2015*; *Buxton et al., 2011*; *Ly et al., 2019*; *Okuda et al., 2002*). No other biological age markers were associated with sex, which contrasts with the study of *Jansen et al., 2021* in adults which reported accelerated biological age in men across all measures tested except for proteomic age. Indeed, the phenomenon of accelerated DNA methylation age in men is well-established (*Horvath and Raj, 2018*), consistent with lower life-expectancies for men. Although it is not known if these biological age differences are due to biological mechanisms or greater prevalence of disease risk factors among men, our data in children before the divergence of risk factor prevalence could indicate a biological mechanism for telomere sex differences and a risk factor-mediated mechanism for other biological age markers. Interestingly, we observed differences in associations between biological age and development between boys and girls, with some consistency across markers: Both shorter telomere length and transcriptome Δ age and were more strongly associated with adiposity in boys, DNA methylation and transcriptome Δ age showed stronger associations among boys with poorer behavior, while in girls both transcriptome and immunometabolic Δ age showed stronger associations with improved attentiveness. Given observed sexual dimorphism in both developmental rates (*León et al., 2014*) and biological age measures through a variety of proposed mechanisms (*Hägg and Jylhävä, 2021*), it may be unsurprising that relationship between biological age and development also differs between the sexes.

Furthermore, we observed that immunometabolic Δ age was associated with greater odds of puberty onset, driven by effects observed among girls only. We did not observe any further significant associations with the onset of puberty, however, the sample size in the subset of children was small compared to the other developmental measures. There was also suggestive evidence for associations between DNA methylation Δ age with the onset of puberty with associations close to the nominal significance threshold. Three previous studies have reported associations between DNA methylation age acceleration and puberty onset and stage (*Binder et al., 2018*; *Simpkin et al., 2017*; *Suarez et al., 2018*), and one study has reported associations between shorter telomere length and puberty onset (*Koss et al., 2020*). However, the directions of effect for telomere length in our study were in the opposite direction. While earlier age at puberty is representative of more advanced physical maturation, it has been associated with metabolic diseases in later life, including cancers (*Cancer", 2012*) and all-cause mortality (*Charalampopoulos et al., 2014*).

We found transcriptome data to be highly accurate in predicting chronological age, including in a test set of children assessed six months later, demonstrating that gene expression tracks closely with age in children, even over this relatively short period. We analyzed biological pathways and processes enriched among transcript clusters contributing to the transcriptome clock, observing the integral role of ribosome and ribosome biogenesis pathways, central to protein synthesis, and biological processes including the immunity-related processes leukocyte migration and activation, and cell movement, activation, and secretion. Strikingly, gene expression in adults is similarly characterized by downregulation of ribosomal genes and enrichment of expression in immune-related genes (*Peters et al., 2015*). This indicates that similar to DNA methylation changes (*Horvath and Raj, 2018*), there is some overlap in gene expression related to both development in children and aging in adults. Although formal testing of enrichment of genes contributing to the transcriptome clock presented here among age-associated genes in adults showed enrichment at only borderline statistical significance, the transcriptome clock predictors are an underrepresentation of the full profile of gene expression associated with age in children, due to the sparsity enforced during the variable selection training process.

Despite associations with growth and adiposity measures, transcriptome age generally showed weaker associations with other developmental outcomes than for the other biological age markers. While this in part can be attributed to the slightly smaller sample size for children with transcriptome data, it is also likely due to the high accuracy in predicting the chronological age of the transcriptome clock, resulting in lower variation in the portion of transcriptome age that is not explained by chronological age, further reducing statistical power. This makes it challenging to judge the relevance of transcriptome age, if any, to developmental endpoints, which may be mixed since a nonsignificant direction of effects were observed with both maturity in attention and lung function, yet relative immaturity in behavior. In fact, training clocks using chronological age, which while providing an accessible route to understanding molecular changes associated with age, does pose limitations

generally for inference regarding biological aging. Particularly for high-dimensional data such as DNA methylation, it has been shown that it is possible to predict chronological age near-perfectly (*Zhang et al., 2019*), thereby limiting information on biological age and its variation. For this reason, newer epigenetic clocks have included clinical and mortality data, to improve clinical relevance and sensitivity to risk factors (*Lu et al., 2019*; *Belsky et al., 2020*), which should be considered in future studies developing clocks in children.

Other limitations include the cross-sectional design of the main analysis, which limits inference regarding the directionality of associations and allows the possibility of age-associated environmental factors to influence the clock development. Furthermore, there were differences in age by study centre. Children from the EDEN cohort study centre were generally older, which likely introduced a degree of cohort bias into the age modeling. For this reason, we adjusted all associations by study centre and additionally assessed age correlation within each study centre. Although cohorts were recruited from the general population, certain ethnicities or socio-economically disadvantaged groups may have been under-represented, limiting generalizability somewhat. A bias towards the over-representation of White ethnic groups is an issue generally with the development of biological clocks, which means associations observed with ethnicity should be interpreted cautiously. While the DNA methylation and transcriptome data were representative of the full genome, our coverage of the metabolome and proteome was limited to targeted assays. For analysis of age at the gene expression and protein/metabolite levels we developed new clocks, and these remain to be validated as true biological age indicators, through testing association with age-related disease and mortality in adults. While biological age markers exist for these data types, they are not appropriate for applying in our dataset, since the same model predictors (i.e. metabolites and proteins) were not included in the assays used here and/or the markers were not trained in pediatric populations, which will drastically reduce their accuracy as molecular changes within in childhood cannot be assumed to follow the same relation with age as in adulthood: For instance, DNA methylation shows a logarithmic dependence with age during childhood and a linear dependence in adulthood (*Horvath, 2013*).

However, the strengths of this study include the large population sample, drawing from six countries from around Europe, increasing generalizability, and the integration of rich molecular data and a broad range of developmental outcomes into a single systematic analysis. Although our age range was somewhat limited, missing the infancy and adolescent periods, the age range covered a key childhood period, where energy expenditure (an indicator of the level of overall physiology) has entered a period of steady increase following more rapid increases during infancy and before stabilization during adolescence (*Pontzer et al., 2021*).

In conclusion, in this large Pan-European study we have found that four indicators of biological age, representing complementary molecular processes, were all associated with BMI after controlling for chronological age, indicating that adiposity is an important correlate of accelerated biological aging in children. We developed a highly accurate 'transcriptome age' clock although it was found to be relatively insensitive to other development phenotypes. We found that immunometabolic Δ age was associated with cognitive maturity fitting a buildup of the biological capital model of aging in children, while shorter telomere length and DNA methylation Δ age was associated with greater behavioral problems suggesting a 'wear and tear' model of aging in children. Our findings contribute to the interpretation and understanding of biological age measures in children, crucial for clinical and epidemiological research into early life risk factors for adverse aging trajectories. Future long-term studies should investigate associations between age acceleration in children and adults to further test the antagonistic pleiotropy hypothesis.

## Acknowledgements

The authors are grateful to all the participating families in the six countries who took part in this study and to all the field workers for their dedication and efficiency. ISGlobal acknowledges support from the Spanish Ministry of Science and Innovation through the 'Centro de Excelencia Severo Ochoa 2019–2023' Program (CEX 2018–000806 S), and support from the Generalitat de Catalunya through the CERCA Programme. The CRG/UPF Proteomics Unit is part of the Spanish Infrastructure for Omics Technologies (ICTS OmicsTech) and it is supported by 'Secretaria d'Universitats i Recerca del Departament d'Economia i Coneixement de la Generalitat de Catalunya' (2017SGR595). We also acknowledge the support of the Spanish Ministry of Science and Innovation to the EMBL partnership, the

Centro de Excelencia Severo Ochoa, and the CERCA Programme/Generalitat de Catalunya. Some figures were created with BioRender.com.

## Additional information

### Funding

| Funder | Grant reference number | Author |
|---|---|---|
| UK Research and Innovation | MR/S03532X/1 | Oliver Robinson |
| European Commission | 308333 874583 | Oliver Robinson<br>ChungHo E Lau<br>Sandra Andrusaityte<br>Eva Borras<br>Paula de Prado-Bert<br>Lida Chatzi<br>Hector C Keun<br>Regina Grazuleviciene<br>Kristine B Gutzkow<br>Lea Maitre<br>Dries S Martens<br>Eduard Sabido<br>Valérie Siroux<br>Jose Urquiza<br>Marina Vafeiadi<br>John Wright<br>Tim S Nawrot<br>Mariona Bustamante |
| Secretaria d'Universitats i Recerca del Departament d'Economia i Coneixement de la Generalitat de Catalunya | 2017SGR595 | Jose Urquiza<br>Lea Maitre<br>Mariona Bustamante<br>Martine Vrijheid |
| Departament de Salut de la Generalitat de Catalunya | SLT017/20/000119 | Jose Urquiza<br>Lea Maitre<br>Mariona Bustamante<br>Martine Vrijheid |

The funders had no role in study design, data collection and interpretation, or the decision to submit the work for publication.

### Author contributions

Oliver Robinson, Conceptualization, Formal analysis, Funding acquisition, Writing - original draft, Project administration, Writing – review and editing; ChungHo E Lau, Formal analysis, Methodology, Writing – review and editing; Sungyeon Joo, Formal analysis, Writing – review and editing; Sandra Andrusaityte, Lea Maitre, Resources, Data curation, Investigation, Writing – review and editing; Eva Borras, Resources, Formal analysis, Writing – review and editing; Paula de Prado-Bert, Valérie Siroux, Methodology, Writing – review and editing; Lida Chatzi, Resources, Funding acquisition, Investigation, Writing – review and editing; Hector C Keun, Eduard Sabido, Resources, Data curation, Writing – review and editing; Regina Grazuleviciene, Resources, Data curation, Funding acquisition, Investigation; Kristine B Gutzkow, John Wright, Resources, Data curation, Funding acquisition, Investigation, Writing – review and editing; Dries S Martens, Resources, Investigation, Methodology, Writing – review and editing; Jose Urquiza, Data curation, Writing – review and editing; Marina Vafeiadi, Resources, Investigation, Writing – review and editing; Tim S Nawrot, Resources, Methodology, Writing – review and editing; Mariona Bustamante, Data curation, Formal analysis, Methodology, Writing – review and editing; Martine Vrijheid, Conceptualization, Resources, Data curation, Funding acquisition, Investigation, Methodology, Writing – review and editing

### Author ORCIDs

Oliver Robinson http://orcid.org/0000-0002-4735-0468
Kristine B Gutzkow http://orcid.org/0000-0002-6716-5921

Eduard Sabido http://orcid.org/0000-0001-6506-7714

### Ethics

Human subjects: Prior to the start of HELIX, all six cohorts had undergone the required evaluation by national ethics committees and obtained all the required permissions for their cohort recruitment and follow-up visits. Each cohort also confirmed that relevant informed consent and approval were in place for secondary use of data from pre-existing data. The work in HELIX was covered by new ethical approvals in each country and at enrolment in the new follow-up, participants were asked to sign a new informed consent form. Additionally, the current study was approved by the Imperial College Research Ethics Committee (Reference: 19IC5567).

### Decision letter and Author response

Decision letter https://doi.org/10.7554/eLife.85104.sa1
Author response https://doi.org/10.7554/eLife.85104.sa2

## Additional files

### Supplementary files

• MDAR checklist

• Source code 1. R script for all data analyses.

• Supplementary file 1. Number of samples and features before and after the quality control process.

• Supplementary file 2. Proportion of covariates missing for each biological age marker.

• Supplementary file 3. Immunometabolic age clock coefficients.

• Supplementary file 4. Tranciptome age clock coefficients.

• Supplementary file 5. Overrepresentation analysis in ConsesuspathDB against KEGG and REACTOME pathways, of all transcripts contributing to the transcriptome clock.

• Supplementary file 6. Overrepresentation analysis in ConsesuspathDB against Gene Ontology (GO) biological process terms, of all transcripts contributing to the transcriptome clock.

• Supplementary file 7. Associations between biological age measures and developmental measures, in main analysis (model 1) and sensitivity analyses (models 2-4).

### Data availability

Due to data protection regulations in each participating country and participant data use agreements, human subject data used in this project cannot be freely shared. The raw data supporting the current study are available on request subject to ethical and legislative review. The "HELIX Data External Data Request Procedures" are available with the data inventory in this website: http://www.projecthelix.eu/data-inventory. The document describes who can apply to the data and how, the timings for approval and the conditions to data access and publication. Researchers who have an interest in using data from this project for reproducibility or in using data held in general in the HELIX data warehouse for research purposes can apply for access to data. Interested researchers should fill in the application protocol found in ANNEX I at https://www.projecthelix.eu/files/helix_external_data_request_proce-dures_final.pdf and send this protocol to helixdata@isglobal.org. The applications are received by the HELIX Coordinator, and are processed and approved by the HELIX Project Executive Committee. All code used for data analysis has been provided as supplementary material. Deidentified dataset for generation of figures 1 and 2 has been provided as a supplementary dataset.

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
