## [Editor Report]

This is an important study that examined multiple biological age measures in children, which has been lacking in literature. The findings of this study provided convincing evidence to interpret and understand the aging and developmental processes in children.

---

## [Decision Letter]

**Decision letter after peer review:**

Thank you for submitting your article "Associations of four biological age markers with child development: A multi-omic analysis in the European HELIX cohort" for consideration by *eLife*. Your article has been reviewed by 2 peer reviewers, and the evaluation has been overseen by a Reviewing Editor and Carlos Isales as the Senior Editor. The reviewers have opted to remain anonymous.

Essential revisions:

1) Pay attention to the reviewer's comments about how to report effects, 95% CIs, and p-values also including multiple testing.

2) Make sure the additional suggestions regarding text are addressed for clarity of the presentation with explanations, terms, research questions, etc.

3) Suggested adjustments and alternative analyses should be addressed.

*Reviewer #1 (Recommendations for the authors):*

1. Introduction-last paragraph: the authors mentioned one of the aims of this study is "to clarify the relationship of age acceleration in children to the buildup of biological capital". It is not clear to me how to define and measure "the buildup of biological capital", can the authors explain more on this aspect?

*Reviewer #2 (Recommendations for the authors):*

General comments:

– Very important aim and many interesting known and novel candidate biological age indicators were used in the analysis. Candidate indicators are an important base for future studies. However, in my opinion, it would have been even more interesting to see how those BA indicators (epigenetic, metabolomic, transcriptomic) that have been developed/established previously in other cohorts (although they are not developed in children) behave in childhood. Consider adding a short explanation in the discussion why the established BA indicators were not and should not be analyzed in populations with this age range.

– When presenting the results and in the discussion, the difference between the BA candidate indicators and established indicators should be clear and highlighted. For example, I do not recommend using the common umbrella term 'biological age indicator' for all four indicators in the discussion. Please, reconsider the conclusions for candidate indicators, because they have not been shown to be true BA indicators.

More specific comments

– In abstract, results:

It is recommended to mention the directions of associations for all relationships that are included in this section. For example, the reader might like to see already in the abstract how height associates with the candidate and known BA indicators. Authors might consider organizing the results into blocks such as 'associations with health risk factors' and 'associations with child developmental measures'. This may help the reader to follow the main conclusions in the abstract.

– In the introduction, the background of the '1st generation clocks' (i.e. clocks built using only calendar age) is introduced but not the 'next generation clocks' (i.e. clocks built using calendar age and some health information). The reader needs to know the difference between these clocks already in the introduction. In addition, the difference between the candidate and the established indicator might be explained.

– In the introduction, page 6: 'To explore these questions, we have performed a comparative analysis…' Please, specify the questions.

– Consider summarizing more information on the different data quality-related reasons behind the data missingness.

To summarize, how many biological samples there were originally, and how many of them per laboratory method (telomere length quantification, DNA methylation, transcriptomics, metabolomics) were excluded because of poor data quality and data analysis-based sex inconsistency?

In addition, how many probes, transcripts, CpG sites, telome length values, and metabolites/biomarkers were excluded due to poor data quality? How many expressed genes there were? These numbers might be added to e.g. existing data flowchart in the supplement or another data flowchart.

– In methods, please, consider rephrasing the sentence where the p-value threshold for significance is explained. Was the threshold in all comparisons multiple testing corrected p-value of 0.05?

– Figure 3 (and other figures showing results from the different subsamples): it is recommended to add sample size (n) (into e.g. title) for the different subsamples in the figure panels to help the reader to remember each indicator was analysed in different subsamples.

---

## [Author Response]

Essential revisions:Reviewer #1 (Recommendations for the authors):1. Introduction-last paragraph: the authors mentioned one of the aims of this study is "to clarify the relationship of age acceleration in children to the buildup of biological capital". It is not clear to me how to define and measure "the buildup of biological capital", can the authors explain more on this aspect?

As we think is explained in the concluding paragraph of the discussion, we interpret associations with positive aspects of development as supporting the “build-up model” and associations with detrimental or less advanced development as supporting a “wear and tear” model. However, we can see this it not apparent in the last paragraph of the introduction. We have therefore altered the introduction as follows:

“We hypothesized that age acceleration would be associated with child development. To test this and assess whether age acceleration is associated with beneficial or detrimental effects on child development, we have performed a comparative analysis of two established and two candidate assessments of biological age within the pan-European Human Early Life Exposome (HELIX) cohort of children aged between 5 and 12 years. We systematically compared associations with developmental endpoints, including growth and adiposity, cognition, behaviour, lung function and pubertal development, and common health risk factors, for telomere length, DNA methylation age, and two newly derived clocks: transcriptome age and immunometabolic age. Through this analysis, we aimed to clarify the interpretation of age acceleration in children, and more broadly develop new biological markers of overall developmental staging in children.”

Reviewer #2 (Recommendations for the authors):General comments:– Very important aim and many interesting known and novel candidate biological age indicators were used in the analysis. Candidate indicators are an important base for future studies. However, in my opinion, it would have been even more interesting to see how those BA indicators (epigenetic, metabolomic, transcriptomic) that have been developed/established previously in other cohorts (although they are not developed in children) behave in childhood. Consider adding a short explanation in the discussion why the established BA indicators were not and should not be analyzed in populations with this age range.

We have added the following sentence:

“For analysis of age at the gene expression and protein/metabolite levels we developed new clocks, and these remain to be validated as true biological age indicators, through testing association with age-related disease and mortality in adults. While biological age markers exist for these data types, they are not appropriate for applying in our dataset, since the same model predictors (i.e., metabolites and proteins) were not included in the assays used here and/or the markers were not trained in pediatric populations, which will drastically reduce their accuracy as molecular changes within in childhood cannot be assumed to follow the same relation with age as in adulthood: For instance, DNA methylation shows a logarithmic dependence with age during childhood and a linear dependence in adulthood [12].”

– When presenting the results and in the discussion, the difference between the BA candidate indicators and established indicators should be clear and highlighted. For example, I do not recommend using the common umbrella term 'biological age indicator' for all four indicators in the discussion. Please, reconsider the conclusions for candidate indicators, because they have not been shown to be true BA indicators.

Thank you for this important observation. However, we still find that “biological age indicator” is a useful umbrella term in this manuscript and there is not an obvious alternative. We therefore have the following sentence on page 8 and highlighted the difference between the markers at key points in the abstract, introduction, results, and discussion.

“We note that since a common definition of markers of biological age is that they should be associated with age-related disease and mortality [69] these new clocks may only currently be considered “candidate” biological age markers. However, we have referred to both the established and candidate markers as biological age markers throughout to simplify presentation.”

More specific comments– In abstract, results:It is recommended to mention the directions of associations for all relationships that are included in this section. For example, the reader might like to see already in the abstract how height associates with the candidate and known BA indicators. Authors might consider organizing the results into blocks such as 'associations with health risk factors' and 'associations with child developmental measures'. This may help the reader to follow the main conclusions in the abstract.

We have revised the Results section of the abstract accordingly.

– In the introduction, the background of the '1st generation clocks' (i.e. clocks built using only calendar age) is introduced but not the 'next generation clocks' (i.e. clocks built using calendar age and some health information). The reader needs to know the difference between these clocks already in the introduction. In addition, the difference between the candidate and the established indicator might be explained.

We have added the following sentence:

“DNA methylation-based clocks, such as the clock of Horvath [12], have been extensively applied in large-scale studies and remain a research field under active development, with “second generation” clocks further incorporating clinical biomarker and mortality information to improve their clinical utility [13, 14].”

– In the introduction, page 6: 'To explore these questions, we have performed a comparative analysis…' Please, specify the questions.

We have altered this paragraph as follows:

“We hypothesized that age acceleration would be associated with child development. To test this and assess whether age acceleration is associated with beneficial or detrimental effects on child development, we have performed a comparative analysis of two established and two candidate assessments of biological age within the pan-European Human Early Life Exposome (HELIX) cohort of children aged between 5 and 12 years.”

– Consider summarizing more information on the different data quality-related reasons behind the data missingness.To summarize, how many biological samples there were originally, and how many of them per laboratory method (telomere length quantification, DNA methylation, transcriptomics, metabolomics) were excluded because of poor data quality and data analysis-based sex inconsistency?In addition, how many probes, transcripts, CpG sites, telome length values, and metabolites/biomarkers were excluded due to poor data quality? How many expressed genes there were? These numbers might be added to e.g. existing data flowchart in the supplement or another data flowchart.

We have added a new supplementary table Supplementary file 1 detailing the QC processes and exclusions for both samples and molecular features. We refer to this table both throughout the molecular data acquisition sections of the methods and in the caption of the existing data flowchart (now figure 1).

– In methods, please, consider rephrasing the sentence where the p-value threshold for significance is explained. Was the threshold in all comparisons multiple testing corrected p-value of 0.05?

We have altered the text as follows to be clearer:

“Due to the exploratory nature of the analysis, we report and discuss associations significant at the both the nominal significance threshold (unadjusted p <0.05) and after correction for 5% false discovery rate using the Benjamini and Hochberg [95] method, calculated across all computed associations (i.e. multiple testing corrected p value <0.05).”

– Figure 3 (and other figures showing results from the different subsamples): it is recommended to add sample size (n) (into e.g. title) for the different subsamples in the figure panels to help the reader to remember each indicator was analysed in different subsamples.

As the numbers in analysis within each subsample also vary depending on outcome, this would require four additional columns in the figure panels which would make the figure too busy. Instead, we added a new table 3, with the sample sizes and point estimates, and also direct readers to the new table 3 in the caption of the figure (now figure 4).